# Scale-Adaptive Balancing of Exploration and Exploitation in Classical Planning

**Stephen Wissow**[*1]**, Masataro Asai**[*2]

[1]University of New Hampshire, [2]MIT-IBM Watson AI Lab, sjw@cs.unh.edu, masataro.asai@ibm.com

## Abstract

Balancing exploration and exploitation has been an important problem in both adversarial games and automated planning. While it has been extensively analyzed in the Multi-Armed Bandit (MAB) literature, and the game community has achieved great success with MAB-based Monte Carlo Tree Search (MCTS) methods, the symbolic planning community has struggled to advance in this area. We describe how Upper Confidence Bound 1's (UCB1's) assumption of reward distributions with known bounded support shared among siblings (arms) is violated when MCTS/Trial-based Heuristic Tree Search (THTS) in previous work uses heuristic values of search nodes in classical planning problems as rewards. To address this issue, we propose a new Gaussian bandit, UCB1-Normal2, and analyze its regret bound. It is variance-aware like UCB1-Normal and UCB-V, but has a distinct advantage: it neither shares UCB-V's assumption of known bounded support nor relies on UCB1-Normal's unfounded conjectures on Student's $t$ and $\chi^2$ distributions. Our theoretical analysis predicts that UCB1-Normal2 will perform well when the estimated variance is accurate, which can be expected in deterministic, discrete, finite state-space search, as in classical planning. Our empirical evaluation confirms that MCTS combined with UCB1-Normal2 outperforms Greedy Best First Search (traditional baseline) as well as MCTS with other bandits.

## 1 Introduction

From the early history of AI and in particular of automated planning and scheduling, heuristic forward search has been a primary methodology for tacking challenging combinatorial problems. A rich variety of search algorithms have been proposed, including Dijkstra search (Dijkstra 1959), A*/ WA* (Hart, Nilsson, and Raphael 1968), and Greedy Best First Search (Bonet and Geffner 2001, GBFS). They are divided into three categories: *optimizing*, which must guarantee the optimality of the output, *satisficing*, which may or may not attempt to minimize solution cost, and *agile*, which ignores solution cost and focuses on finding a solution quickly. This paper focuses on the agile setting.

Unlike optimizing search, theoretical understanding of satisficing and agile search has been limited. Recent theoretical work on GBFS (Heusner, Keller, and Helmert 2017,

2018b,a; Kuroiwa and Beck 2022) refined the concept of search progress in agile search, but only based on a post hoc analysis that depends on oracular information, making their insights difficult to apply to practical search algorithm design, although it has been recently applied to a learning-based approach (Ferber et al. 2022a). More importantly, their analysis is incompatible with a wider range of randomized algorithms (Nakhost and Müller 2009; Imai and Kishimoto 2011; Kishimoto, Zhou, and Imai 2012; Valenzano et al. 2014; Xie, Nakhost, and Müller 2012; Xie, Müller, and Holte 2014; Xie et al. 2014; Xie, Müller, and Holte 2015; Asai and Fukunaga 2017; Kuroiwa and Beck 2022) that outperform the deterministic baseline with randomized explorations; as a result, their detailed theoretical properties are largely unknown except for probabilistic completeness (Valenzano et al. 2014). It is unsurprising that analyzing randomized algorithms requires a statistical perspective, which is also growing more important due to recent advances in learned heuristic functions (Toyer et al. 2018; Ferber, Helmert, and Hoffmann 2020; Shen, Trevizan, and Thiébaux 2020; Ferber et al. 2022b; Rivlin, Hazan, and Karpas 2019; Gehring et al. 2022; Garrett, Kaelbling, and Lozano-Pérez 2016).

In this paper, we tackle the problem of balancing exploration and exploitation in classical planning through a statistical lens and from the perspective of MABs. Previous work showed that traditional forward search algorithms (A*, GBFS) can be seen as a form of MCTS, but we refine and recast this paradigm as a repeated process of collecting a reward dataset and exploring the environment based on estimates obtained from this dataset. This perspective reveals a theoretical issue in THTS (Schulte and Keller 2014), a MCTS modified for classical planning that uses UCB1: the optimization objective of classical planning has no a priori known bound, and this violates the bounded reward assumption of UCB1.

To apply MAB to classical planning without THTS's theoretical issues, we propose UCB1-Normal2, a new Gaussian bandit, and GreedyUCT-Normal2, a new agile planning algorithm that combines MCTS with UCB1-Normal2, and show that GreedyUCT-Normal2 outperforms traditional agile algorithms (GBFS), existing MCTS-based algorithms (GreedyUCT, GreedyUCT*), and other MCTS-based algorithms combined with existing variance-aware bandits

---

[*]These authors contributed equally.

(UCB1-Normal and UCB-V).

In summary, our core contributions are as follows.

- We identify theoretical issues that arise when applying UCB1 to planning tasks.

- To address these issues, we present UCB1-Normal2, a new Gaussian bandit. We analyze its regret bound, which improves as the estimated variance is closer to the true variance, and is constant when they match. This makes it particularly powerful in a deterministic and finite state space such as classical planning.

- We present GreedyUCT-Normal2, a new forward search algorithm that combines UCB1-Normal2 with MCTS and outperforms existing algorithms in agile classical planning.

## 2 Background

### 2.1 Classical Planning

We define a propositional STRIPS Planning problem as a 4-tuple $[P, A, I, G]$ where $P$ is a set of propositional variables, $A$ is a set of actions, $I \subseteq P$ is the initial state, and $G \subseteq P$ is a goal condition. Each action $a \in A$ is a 4-tuple $[\text{PRE}(a), \text{ADD}(a), \text{DEL}(a), \text{C}(a)]$ where $\text{C}(a) \in \mathbb{Z}^{0+}$ is a cost, $\text{PRE}(a) \subseteq P$ is a precondition and $\text{ADD}(a), \text{DEL}(a) \subseteq P$ are the add-effects and delete-effects. A state $s \subseteq P$ is a set of true propositions (all of $P \setminus s$ is false), an action $a$ is *applicable* when $s \supseteq \text{PRE}(a)$ (read: $s$ *satisfies* $\text{PRE}(a)$), and applying action $a$ to $s$ yields a new successor state $a(s) = (s \setminus \text{DEL}(a)) \cup \text{ADD}(a)$.

The task of classical planning is to find a sequence of actions called a *plan* $(a_1, \cdots, a_n)$ where, for $1 \leq t \leq n$, $s_0 = I$, $s_t \supseteq \text{PRE}(a_{t+1})$, $s_{t+1} = a_{t+1}(s_t)$, and $s_n \supseteq G$. A plan is *optimal* if there is no plan with lower cost $\sum_t \text{C}(a_t)$. A plan is otherwise called *satisficing*. In this paper, we assume unit-cost: $\forall a \in A; \text{C}(a) = 1$.

A domain-independent heuristic function $h$ in classical planning is a function of a state $s$ and the problem $[P, A, I, G]$, but the notation $h(s)$ usually omits the latter. It returns an estimate of the cumulative cost from $s$ to one of the goal states (which satisfy $G$), typically through a symbolic, non-statistical means including problem relaxation and abstraction. Notable state-of-the-art functions that appear in this paper include $h^{\text{FF}}, h^{\max}, h^{\text{add}}$, and $h^{\text{GC}}$ (Hoffmann and Nebel 2001; Bonet and Geffner 2001; Fikes, Hart, and Nilsson 1972). Their implementation details are beyond the scope of this paper, and are included in the appendix Sec. S1.

### 2.2 Multi-Armed Bandit (MAB)

MAB (Thompson 1933; Robbins 1952; Bush and Mosteller 1953) is a problem of finding the best strategy to choose from multiple unknown reward distributions. It is typically depicted by a row of $K$ slot machines each with a lever or "arm." Each time the player plays one of the machines and pulls an arm (a *trial*), the player receives a reward sampled from the distribution assigned to that arm. Through multiple trials, the player discovers the arms' distributions and selects arms to maximize the reward.

The most common optimization objective of MAB is *Cumulative Regret* (CR) minimization. Let $r_i$ ($1 \leq i \leq K$) be a random variable (RV) for the reward that we would receive when we pull arm $i$. We call $p(r_i)$ an unknown *reward distribution* of $i$. Let $t_i$ be a RV of the number of trials performed on arm $i$ and $T = \sum_i t_i$ be the total number of trials across all arms.

**Definition 1.** *The* cumulative regret $\Delta$ *is the gap between the optimal and the actual expected cumulative reward:* $\Delta = T \max_i \mathbb{E}[r_i] - \sum_i \mathbb{E}[t_i]\mathbb{E}[r_i]$.

Algorithms whose regret per trial $\Delta/T$ converges to 0 with $T \to \infty$ are called *zero-regret*. Those with a logarithmically upper-bounded regret, $O(\log T)$, are also called *asymptotically optimal* because this is the theoretical optimum achievable by any algorithm (Lai, Robbins et al. 1985).

Upper Confidence Bound 1 (Auer, Cesa-Bianchi, and Fischer 2002, UCB1) is a logarithmic CR MAB for rewards $r_i \in [0, c]$ with known $c$. Let $r_{i1} \ldots r_{it_i} \sim p(r_i)$ be $t_i$ i.i.d. samples obtained from arm $i$. Let $\hat{\mu}_i = \frac{1}{t_i} \sum_{j=1}^{t_i} r_{ij}$. To minimize CR, UCB1 selects $i$ with the largest Upper Confidence Bound defined below.

$$\begin{aligned} \text{UCB1}_i &= \hat{\mu}_i + c\sqrt{2 \log T / t_i} \\ \text{LCB1}_i &= \hat{\mu}_i - c\sqrt{2 \log T / t_i} \end{aligned} \tag{1}$$

For reward (cost) minimization, LCB1 instead select $i$ with the smallest Lower Confidence Bound defined above (e.g., in Kishimoto et al. (2022)), but we may use the terms U/LCB1 interchangeably. UCB1's second term is often called an *exploration term*. Generally, an LCB is obtained by flipping the sign of the exploration term in a UCB. U/LCB1 refers to a specific algorithm while U/LCB refers to general confidence bounds. $c$ is sometimes set heuristically as a hyperparameter called the *exploration rate*.

### 2.3 Forward Heuristic Best-First Search

Classical planning problems are typically solved as a path finding problem defined over a state space graph induced by the transition rules, and the current dominant approach is based on *forward search*. Forward search maintains a set of search nodes called an *open list*. They repeatedly (1) (*selection*) select a node from the open list, (2) (*expansion*) generate its successor nodes, (3) (*evaluation*) evaluate the successor nodes, and (4) (*queueing*) reinsert them into the open list. Termination typically occurs when a node is expanded that satisfies a goal condition, but a satisficing/agile algorithm can perform *early goal detection*, which immediately checks whether any successor node generated in step (2) satisfies the goal condition. Since this paper focuses on agile search, we use early goal detection for all algorithms.

Within forward search, forward *best-first* search defines a particular ordering in the open list by defining *node evaluation criteria* (NEC) $f$ for selecting the best node in each iteration. Let us denote a node by $n$ and the state represented by $n$ as $s_n$. As NEC, Dijkstra search uses $f_{\text{Dijkstra}}(n) = g(n)$ ($g$-value), the minimum cost from the initial state $I$ to the state $s_n$ found so far. $A^*$ uses $f_{A^*}(n) = g(n) + h(s_n)$, the sum of $g$-value and the value returned by a heuristic function

$h$ ($h$-value). GBFS uses $f_{\text{GBFS}}(n) = h(s_n)$. Forward best-first search that uses $h$ is called forward *heuristic* best-first search. Dijkstra search is a special case of A* with $h(s) = 0$.

Typically, an open list is implemented as a priority queue ordered by NEC. Since the NEC can be stateful, e.g., $g(s_n)$ can update its value, a priority queue-based open list assumes monotonic updates to the NEC because it has an unfavorable time complexity for removals. A*, Dijkstra, and GBFS satisfy this condition because $g(n)$ decreases monotonically and $h(s_n)$ is constant.

MCTS is a class of forward heuristic best-first search that represents the open list as the leaves of a tree. We call the tree a *tree-based open list*. Our MCTS is based on the description in Keller and Helmert (2013) and Schulte and Keller (2014), whose implementation details are available in the appendix (Sec. S2). Overall, MCTS works in the same manner as other best-first search with a few key differences. (1) (*selection*) To select a node from the tree-based open list, it recursively selects an action on each branch of the tree, start from the root, using the NEC to select a successor node, descending until reaching a leaf node. (Sometimes the action selection rule is also called a *tree policy*.) At the leaf, it (2) (*expansion*) generates successor nodes, (3) (*evaluation*) evaluates the new successor nodes, (4) (*queueing*) attaches them to the leaf, and *backpropagates* (or *backs-up*) the information to the leaf's ancestors, all the way up to the root.

The evaluation obtains a heuristic value $h(s_n)$ of a leaf node $n$. In adversarial games like Backgammon or Go, it is obtained either by (1) hand-crafted heuristics, (2) *playouts* (or *rollouts*) where the behaviors of both players are simulated by uniformly random actions (*default policy*) until the game terminates, or (3) a hybrid *truncated simulation*, which returns a hand-crafted heuristic after performing a short simulation (Gelly and Silver 2011). In recent work, the default policy is replaced by a learned policy (Silver et al. 2016).

Trial-based Heuristic Tree Search (Keller and Helmert 2013; Schulte and Keller 2014, THTS), a MCTS for classical planning, is based on two key observations: (1) the rollout is unlikely to terminate in classical planning due to sparse goals, unlike adversarial games, like Go, which are guaranteed to finish in a limited number of steps with a clear outcome (win/loss); and (2) a tree-based open list can reorder nodes efficiently under non-monotonic updates to NEC, and thus is more flexible than a priority queue-based open list, and can readily implement standard search algorithms such as A* and GBFS without significant performance penalty. We no longer distinguish THTS and MCTS and imply that the former is included in the latter, because THTS is a special case of MCTS with an immediately truncated default policy simulation.

Finally, Upper Confidence Bound applied to trees (Kocsis and Szepesvári 2006, UCT) is a MCTS that uses UCB1 for action selection and became widely popular in adversarial games. Schulte and Keller (2014) proposed several variants of UCT including GreedyUCT (GUCT), UCT*, and GreedyUCT* (GUCT*). We often abbreviate a set of algorithms to save space, e.g., [G]UCT[*] denotes $\{\text{UCT}, \text{GUCT}, \text{UCT}^*, \text{GUCT}^*\}$. In this paper, we mainly discuss GUCT[*] due to our focus on the agile satisficing setting that does not prioritize minimization of solution cost.

## 3 Theoretical Issues in Existing MCTS-based Classical Planning

We revisit A* and GBFS implemented as MCTS from a statistical perspective. Let $S(n)$ be the set of successors of a node $n$, $L(n)$ be the set of leaf nodes in the subtree under $n$, and $\text{c}(n, n')$ be the path cost between $n$ and $n'$ on the tree (equivalent to an action cost if $n'$ is a successor of $n$). We define the NECs of A* and GBFS as $f_{A^*}(n) = g(n) + h_{A^*}(n)$ and $f_{\text{GBFS}}(n) = h_{\text{GBFS}}(n)$ which satisfy the following equations, shown by expanding $h_{A^*}$ and $h_{\text{GBFS}}$ recursively and assuming $h_{A^*}(n') = h_{\text{GBFS}}(n') = h(s_{n'})$ if $n'$ is a leaf.

$$h_{A^*}(n) = \min_{n' \in S(n)}[\text{c}(n, n') + h_{A^*}(n')]$$
$$= \min_{n' \in L(n)}[\text{c}(n, n') + h(s_{n'})]$$
$$h_{\text{GBFS}}(n) = \min_{n' \in S(n)}[h_{\text{GBFS}}(n')]$$
$$= \min_{n' \in L(n)}[h(s_{n'})]$$

Observe that these NECs estimate the minimum of the cost-to-go from the *dataset/samples* $L(n)$. The minimum is also known as an *order statistic*; other order statistics include the top-$k$ element, the $q$-quantile, and the median (0.5-quantile). In contrast, [G]UCT computes the average (instead of minimum) over the dataset, and adds an exploration term to the average based on LCB1:

$$h_{\text{UCT}}(n) = \frac{1}{|L(n)|} \sum_{n' \in S(n)} |L(n')|(\text{c}(n, n')$$
$$+ h_{\text{UCT}}(n'))$$
$$= \frac{1}{|L(n)|} \sum_{n' \in L(n)} (\text{c}(n, n') + h(s_{n'}))$$
$$h_{\text{GUCT}}(n) = \frac{1}{|L(n)|} \sum_{n' \in S(n)} |L(n')| h_{\text{GUCT}}(n')$$
$$= \frac{1}{|L(n)|} \sum_{n' \in L(n)} h(s_{n'})$$
$$f_{\text{UCT}}(n) = g(n) + h_{\text{UCT}}(n) - c\sqrt{(2 \log |L(p)|)/|L(n)|}$$
$$f_{\text{GUCT}}(n) = h_{\text{GUCT}}(n) - c\sqrt{(2 \log |L(p)|)/|L(n)|}$$

where $p$ is a parent node of $n$ and $|L(p)|$ is the number of leaf nodes in the subtree of the parent. $|L(p)|$ and $|L(n)|$ respectively correspond to $T$ and $t_i$ in Eq. 1. Note that the term "monte-carlo estimate" is commonly used in the context of estimating the integral/expectation/average, but less often in estimating the maximum/minimum, though we continue using the term MCTS.

From the statistical estimation standpoint, existing MCTS-based planning algorithms have a number of theoretical issues. First, note that the samples of heuristic values collected from $L(n)$ correspond to the rewards in the MAB algorithms, and that UCB1 assumes reward distributions with *known bounds shared by all arms*. However, such *a priori* known bounds do not exist for the heuristic values of classical planning, unlike adversarial games whose rewards are either +1/0 or +1/-1 representing a win/loss. Also, usually the range of heuristic values in each subtree of the search tree substantially differ from each other. Schulte and

Keller (2014) claimed to have addressed this issue by modifying the UCB1, but their modification does not fully address the issue, as we discuss below.

$$f_{\text{GUCT-01}}(n)$$
$$= \frac{h_{\text{GUCT}}(n)-m}{M-m} - c\sqrt{(2\log|L(p)|)/|L(n)|} \quad (2)$$
$$m + (M-m)f_{\text{GUCT-01}}(n)$$
$$= h_{\text{GUCT}}(n) - c(M-m)\sqrt{(2\log|L(p)|)/|L(n)|} \quad (3)$$

Let us call their variant *GUCT-01*. GUCT-01 normalizes the first term of the NEC to $[0,1]$ by taking the minimum and maximum among $n$'s siblings sharing the parent $p$. Given $M = \max_{n' \in S(p)} h_{\text{GUCT}}(n')$, $m = \min_{n' \in S(p)} h_{\text{GUCT}}(n')$, and a hyperparameter $c$, GUCT-01 modifies $f_{\text{GUCT}}$ into $f_{\text{GUCT-01}}$ (Eq. 2). However, the node ordering by NEC is maintained when all arms are shifted and scaled by the same amount, thus GUCT-01 is identical to the standard UCB1 with a reward range $[0, c(M-m)]$ for all arms (Eq. 3); we additionally note that this version avoids a division-by-zero issue for $M-m=0$.

There are two issues in GUCT-01: First, GUCT-01 does not address the fact that different subtrees have different ranges of heuristic values. Second, we would expect GUCT-01 to explore excessively, because the range $[0, c(M-n)]$ obtained from the data of the entire subtree of the parent is always broader than that of each child, since the parent's data is a union of those from all children. We do note that $M-m$ differs for each parent, and thus GUCT-01 adjusts its exploration rate in a different parts of the search tree. In other words, GUCT-01 is depth-aware, but is not breadth-aware: it considers the reward range only for the parent, and not for each child.

Further, in an attempt to improve the performance of [G]UCT, Schulte and Keller (2014) noted that using the average is "rather odd" for planning, and proposed UCT* and GreedyUCT* (GUCT*) which combines $h_{A^*}$ and $h_{\text{GBFS}}$ with LCB1 without statistical justification.

Finally, these variants failed to improve over traditional algorithms (e.g., GBFS) unless combined with various other enhancements such as deferred heuristic evaluation (DE) and preferred operators (PO). The theoretical characteristics of these enhancements are not well understood, rendering their use ad hoc and the reason for GUCT-01's performance inconclusive, and motivating better theoretical analysis.

## 4 Bandit Algorithms with Unbounded Distributions with Different Scales

To handle reward distributions with unknown support that differs across arms, we need a MAB that assumes an unbounded reward distribution spanning the real numbers. We use the Gaussian distribution here, although future work may consider other distributions. Formally, we assume each arm $i$ has a reward distribution $\mathcal{N}(\mu_i, \sigma_i^2)$ for some unknown $\mu_i, \sigma_i^2$. As $\sigma_i^2$ differs across $i$, the reward uncertainty differs across the arms. By contrast, the reward uncertainty of each arm in UCB1 is expressed by the range $[0, c]$, which is the same across the arms. We now discuss the shortcomings of

MABs from previous work (Eq. 4-7), and present our new MAB (Eq. 8).

$$\text{UCB1-Normal}_i = \hat{\mu}_i + \hat{\sigma}_i\sqrt{(16\log T)/t_i} \quad (4)$$
$$\text{UCB1-Tuned}_1 = \quad (5)$$
$$\hat{\mu}_i + c\sqrt{\min(1/4, \hat{\sigma}_i^2 + \sqrt{2\log T/t_i})\log T/t_i}$$
$$\text{UCB-V}_i = \hat{\mu}_i + \hat{\sigma}_i\sqrt{(2\log T)/t_i} + (3c\log T)/t_i \quad (6)$$
$$\text{Bayes-UCT2}_i = \hat{\mu}_i^{\text{Bayes}} + \hat{\sigma}_i^{\text{Bayes}}\sqrt{2\log T} \quad (7)$$
$$\text{UCB1-Normal2}_i = \hat{\mu}_i + \hat{\sigma}_i\sqrt{2\log T} \quad (8)$$

The UCB1-Normal MAB (Auer, Cesa-Bianchi, and Fischer 2002, Theorem 4), which was proposed along with UCB1 [idem, Theorem 1], is designed exactly for this scenario but is still unpopular. Given $t_i$ i.i.d. samples $r_{i1}\ldots r_{it_i} \sim \mathcal{N}(\mu_i, \sigma_i^2)$ from each arm $i$ where $T = \sum_i t_i$, it chooses $i$ that maximizes the metric shown in Eq. 4. To apply this bandit to MCTS, substitute $T = |L(p)|$ and $t_i = |L(n)|$, and backpropagate the statistics $\hat{\mu}_i, \hat{\sigma}_i^2$ (see Appendix Sec. S4). For minimization tasks such as classical planning, use the LCB. We refer to the GUCT variant using UCB1-Normal as *GUCT-Normal*. An advantage of UCB1-Normal is its logarithmic upper bound on regret (Auer, Cesa-Bianchi, and Fischer 2002, Appendix B). However, it did not perform well in our empirical evaluation, likely because its proof relies on two conjectures which are explicitly stated by the authors as not guaranteed to hold.

**Theorem 1** (From (Auer, Cesa-Bianchi, and Fischer 2002)). *UCB1-Normal has a logarithmic regret-per-arm* $256\frac{\sigma_i^2 \log T}{\Delta_i^2} + 1 + \frac{\pi^2}{2} + 8\log T$ *if, for a Student's $t$ RV $X$ with* $s$ *degrees of freedom (DOF),* $\forall a \in [0, \sqrt{2(s+1)}]; P(X \geq a) \leq e^{-a^2/4}$, *and if, for a $\chi^2$ RV $X$ with $s$ DOF, $P(X \geq 4s) \leq e^{-(s+1)/2}$.*

To avoid relying on these two conjectures, we need an alternate MAB that similarly adjusts the exploration rate based on the variance. Candidates include UCB1-Tuned (Auer, Cesa-Bianchi, and Fischer 2002) in Eq. 5, UCB-V (Audibert, Munos, and Szepesvári 2009) in Eq. 6, and Bayes-UCT2 (Tesauro, Rajan, and Segal 2010) in Eq. 7 (not to be confused with Bayes-UCB (Kaufmann, Cappé, and Garivier 2012)), but they all have various limitations. UCB1-Tuned assumes a bounded reward distribution and lacks a regret bound. UCB-V improves UCB1-Tuned with a regret proof but it also assumes a bounded reward distribution. Bayes-UCT2 lacks a regret bound, proves its convergence only for bounded reward distributions, lacks a thorough ablation study for its 3 modifications to UCB1-based MCTS, and lacks evaluation on diverse tasks as it is tested only on a synthetic tree (fixed depth, width, and rewards).

We present UCB1-Normal2 (Eq. 8), a new, conservative, trimmed-down version of Bayes-UCT2, and analyze its regret bound.

**Theorem 2** (Main Result). *Let $\alpha \in [0,1]$ be an unknown problem-dependent constant and $\chi^2_{1-\alpha,n}$ be the critical*

*value for the tail probability of a $\chi^2$ distribution with significance $\alpha$ and DOF $n$ that satisfies $P(t_i\hat{\sigma}_i^2/\sigma_i^2 < \chi_{1-\alpha,t_i}^2) = \alpha$. UCB1-Normal2 has a worst-case polynomial, best-case constant regret-per-arm*

$$\frac{-4(\log\alpha)\sigma_i^2\log T}{\Delta_i^2} + 1 + 2C + \frac{(1-\alpha)T(T+1)(2T+1)}{3}$$

$$\xrightarrow{\alpha\to 1} 1 + 2C$$

*where $C$ is a finite constant if each arm is pulled $M = \inf\{n | 8 < \chi_{1-\alpha,n}^2\}$ times in the beginning.*

*Proof. (Sketch of appendix Sec. S3.2-S3.3.)* We use Hoeffding's inequality for sub-Gaussian distributions as Gaussian distributions belong to sub-Gaussian distributions. Unlike in UCB1 where the rewards have a fixed known support $[0, c]$, we do not know the true reward variance $\sigma_i^2$. Therefore, we use the fact that $t_i\hat{\sigma}_i{}^2/\sigma_i^2$ follows a $\chi^2$ distribution and $P(t_i\hat{\sigma}_i{}^2/\sigma_i^2 < \chi_{1-\alpha,t_i}^2) = \alpha$ for some $\alpha$. We use union-bound to address the correlation and further upper-bound the tail probability. We also use $\chi_{1-\alpha,t_i}^2 \geq \chi_{1-\alpha,2}^2 = -2\log\alpha$ for $t_i \geq 2$. The resulting upper bound contains an infinite series $C$. Its convergence condition dictates the minimum pulls $M$ that must be performed initially. $\square$

Polynomial regrets are generally worse than logarithmic regrets of UCB1-Normal. However, our regret bound improves over that of UCB1-Normal if $T$ is small and $\alpha \approx 1$ ($\log\alpha \approx 0, 1-\alpha \approx 0$). $\alpha$ represents the accuracy of the sample variance $\hat{\sigma}^2$ toward the true variance $\sigma^2$. In deterministic, discrete, finite state-space search problems like classical planning, $\alpha$ tends to be close to (or sometimes even match) 1 because $\sigma = \hat{\sigma}$ is achievable. Several factors of classical planning contribute to this. Heuristic functions in classical planning are deterministic, unlike rollout-based heuristics in adversarial games. This means $\sigma = \hat{\sigma} = 0$ when a subtree is linear due to the graph shape. Also, $\sigma = \hat{\sigma}$ when all reachable states from a node are exhaustively enumerated in its subtree. In statistical terms, this is because draws from heuristic samples are performed without replacements due to duplication checking.

Unlike UCB-V and UCB1-Normal, our MCTS+UCB1-Normal2 algorithm does not need explicit initialization pulls because every node is evaluated once and its heuristic value is used as a single sample. This means we assume $M = 1$, thus $\alpha > \text{ERF}(2) > 0.995$ because $8 < \chi_{1-\alpha,1}^2 \Leftrightarrow 1 - \alpha < \frac{\gamma(\frac{1}{2},\frac{8}{2})}{\Gamma(\frac{1}{2})} = 1 - \text{ERF}(2)$. In classical planning, this assumption is more realistic than the conjectures used by UCB1-Normal.

## 5 Experimental Evaluation

We evaluated the efficiency of our algorithms in terms of the number of nodes evaluated before a goal is found. We used a python-based implementation (Alkhazraji et al. 2020, Pyperplan) for convenient prototyping. It is slower than C++-based state-of-the-art systems (e.g. Fast Downward (Helmert 2006)), but our focus on evaluations makes this irrelevant and also improves reproducibility by avoiding the effect of hardware differences and low-level implementation details.

We evaluated the algorithms over a subset of the International Planning Competition benchmark domains,[1] selected for compatibility with the set of PDDL extensions supported by Pyperplan. The program terminates either when it reaches 10,000 node evaluations or when it finds a goal. In order to limit the length of the experiment, we also had to terminate the program on problem instances whose grounding took more than 5 minutes. The grounding limit removed 113 instances from freecell, pipesworld-tankage, and logistics98. This resulted in 751 problem instances across 24 domains in total. We evaluated various algorithms with $h^{\text{FF}}$, $h^{\text{add}}$, $h^{\text{max}}$, and $h^{\text{GC}}$ (goal count) heuristics (Fikes, Hart, and Nilsson 1972), and our analysis focuses on $h^{\text{FF}}$. We included $h^{\text{GC}}$ because it can be used in environments without domain descriptions, e.g., in the planning-based approach (Lipovetzky, Ramírez, and Geffner 2015) to the Atari environment (Bellemare et al. 2015). We ran each configuration with 5 random seeds and report the average number of problem instances solved. To see the spread due to the seeds, see the cumulative histogram plots Fig. S1-S3 in the appendix.

We evaluated the following algorithms: **GBFS** is GBFS implemented on priority queue. **GUCT** is a GUCT based on the original UCB1. **GUCT-01** is GUCT with ad hoc $[0, 1]$ normalization of the mean (Schulte and Keller 2014). **GUCT-Normal/-Normal2/-V** are GUCT variants using UCB1-Normal/UCB1-Normal2/UCB-V respectively. The starred variants **GUCT*/-01/-Normal/-Normal2** are using $h_{\text{GBFS}}$ backpropagation (Schulte and Keller 2014, called *full-bellman backup*). For GUCT and GUCT-01, we evaluated the hyperparameter $c$ with the standard value $c = 1.0$ and $c = 0.5$. The choice of the latter is due to Schulte and Keller (2014), who claimed that GUCT [*]-01 performed the best when $0.6 < C = c\sqrt{2} < 0.9$, i.e., $0.4 < c < 0.63$. Our aim of testing these hyperparameters is to compare them against automatic exploration rate adjustments performed by UCB1-Normal[2].

Schulte and Keller (2014) previously reported that two ad hoc enhancements to GBFS, PO and DE, also improve the performance of GUCT [*]-01. We implemented them in our code, and show the results. We do not report configurations unsupported by the base Pyperplan system: GBFS+PO, and PO with heuristics other than $h^{\text{FF}}$.

**Reproduction and a More Detailed Ablation of Previous Work** We first reproduced the results in (Schulte and Keller 2014) and provides its more detailed ablation. Table 1 shows that GUCT [*][-01] is indeed significantly outperformed by the more traditional algorithm GBFS, indicating that UCB1-based exploration is not beneficial for planning. Although this result disagrees with the final conclusion of their paper, their conclusion relied on incorporating the DE and PO enhancements, and these confounding factors impede conclusive analysis.

Our ablation includes the effect of min-/max-based mean normalization (Eq. 2), which was not previously evaluated. GUCT [*]-01 performs significantly worse than GUCT [*] which has no normalization. This implies that normalization

---

[1]github.com/aibasel/downward-benchmarks

| $h =$ | $h^{\mathrm{FF}}$ | | $h^{\mathrm{add}}$ | | $h^{\mathrm{max}}$ | | $h^{\mathrm{GC}}$ | | $h^{\mathrm{FF}}$+PO | | $h^{\mathrm{FF}}$+DE | | $h^{\mathrm{FF}}$+DE+PO | |
|---|---|---|---|---|---|---|---|---|---|---|---|---|---|---|
| $c =$ | 0.5 | 1 | 0.5 | 1 | 0.5 | 1 | 0.5 | 1 | 0.5 | 1 | 0.5 | 1 | 0.5 | 1 |
| GUCT | 413.2 | 396.4 | 405.8 | 373.8 | 224.8 | 222.2 | 296 | 278 | 439.2 | 411.8 | 418.6 | 354.6 | 450 | 393.2 |
| * | 508.8 | 440.8 | 496.2 | 453.8 | 239.4 | 234.2 | 306.2 | 303 | 542.4 | 448 | 441.8 | 386.8 | 477 | 422 |
| -01 | 369.6 | 354.8 | 345.2 | 312.8 | 242.2 | 227.6 | 307 | 295.2 | 403.2 | 387 | 355.6 | 344.8 | 406.4 | 404.4 |
| *-01 | 393.6 | 372 | 373 | 343.6 | 236.2 | 226.4 | 306.2 | 289.8 | 430.2 | 401.2 | 377.6 | 363 | 426.2 | 421.2 |
| -V | 329.8 | 307.2 | 325 | 297.6 | 215 | 200 | 264.8 | 243.8 | 383.8 | 348.4 | 334.4 | 310 | 384.4 | 377.4 |
| -Normal | - | 278 | - | 261.4 | - | 209.2 | - | 231.8 | - | 331.6 | - | 269.2 | - | 342.6 |
| *-Normal | - | 311.6 | - | 294.8 | - | 212.2 | - | 244 | - | 338.2 | - | 285.2 | - | 343.8 |
| -Normal2 | - | **563.8** | - | **519.2** | - | **301** | - | **374.6** | - | **596.4** | - | **496.8** | - | **550.8** |
| *-Normal2 | - | **551.2** | - | **516.2** | - | **258.2** | - | 338.6 | - | **593.8** | - | **490.6** | - | **543.4** |
| GBFS | - | 522.4 | - | 501.6 | - | 221.4 | - | **351.2** | - | - | - | 474 | - | - |

Table 1: The number of problem instances solved with less than 10,000 node evaluations; top two configurations in **bold** for each heuristic; each number represents an average over 5 trials. We show results for both $c = 1.0$ and $c = 0.5$ ("best parameter" according to Schulte and Keller (2014)) when the algorithm requires one. Algorithms in the bottom half have no hyperparameter. PO and DE stand for Preferred Operators and Deferred Evaluation. It does not contain PO for GBFS and heuristics other than $h^{\mathrm{FF}}$ due to the lack of support in Pyperplan.

in GUCT [*]-01 not only failed to address the theoretical issue of applying UCB1 to rewards with unknown and different supports, but also had an adverse effect on node evaluations due to the excessive exploration, as predicted by our analysis in Sec. 3.

**The Effect of Scale Adaptability** We compare the performance of various algorithms in terms of the number of problem instances solved. First, GUCT-Normal2 outperforms GBFS, making it the first instance of MCTS that performs better than traditional algorithms by its own (without various other enhancements). Overall, GUCT-Normal2 performed well with all 4 heuristics.

GUCT-Normal2 also significantly outperformed GUCT/GUCT-01/-Normal/-V and their GUCT* variants. The dominance against GUCT-Normal is notable because this supports our analysis that in classical planning $\hat{\sigma}^2 \approx \sigma^2$, thus $P(t_i\hat{\sigma}^2/\sigma^2 < \chi^2_{1-\alpha,t_i}) = \alpha \approx 1$, overcoming the asymptotic deficit (the polynomial regret in GUCT-Normal2 vs. the logarithmic regret of GUCT-Normal).

While the starred variants (GUCT*, etc) can be significantly better than the non-starred variants (GUCT) at times, this trend was opposite in algorithms that perform better, e.g., GUCT*-Normal2 tend to be worse than GUCT-Normal2. This supports our claim that Full-Bellman backup proposed by (Schulte and Keller 2014) is theoretically unfounded and thus does not consistently improve search algorithms. Further theoretical investigation of a similar maximum-based backup is an important avenue of future work.

The table also compares GUCT [*]-Normal[2], which do not require any hyperparameter, against GUCT [*][-01/-V] with different $c$ values. Although $c = 0.5$ improves the performance of GUCT [*]-01 as reported by (Schulte and Keller 2014), it did not improve enough to catch up with the adaptive exploration rate adjustment of GUCT [*]-Normal2. We tested a larger variety of $c$-values and did not observe significant change.

**Preferred Operators** Some heuristic functions based on problem relaxation, notably $h^{\mathrm{FF}}$, compute a solution of the delete-relaxed problem, called a relaxed plan, and return its cost as the heuristic value (see appendix Sec. S1 for details). Actions included in a relaxed plan are called "helpful actions" (Hoffmann and Nebel 2001) or "preferred operators" (Richter and Helmert 2009) and are used by a planner in a variety of ways (e.g., initial incomplete search of FF planner (Hoffmann and Nebel 2001) and alternating open list in LAMA planner (Richter, Westphal, and Helmert 2011)). Schulte and Keller (2014) used it in MCTS/THTS by limiting the action selection to the preferred operators, and falling back to original behavior if no successors qualify. In MCTS terminology (Sec. 2.3), this is a way to modify the tree policy by re-weighting with a mask. We reimplemented the same strategy in our code base. Our result shows that it also improves GUCT [*][-Normal2], consistent with the improvement in GUCT [*]-01 previously reported.

**Deferred Heuristic Evaluation** Table 1 shows the effect of deferred heuristic evaluation (DE) on search algorithms. In this experiment, DE is expected to degrade the number compared to the algorithms with eager evaluations because deferred evaluation trades the number of calls to heuristics with the number of nodes inserted to the tree, which is limited to 10,000. When CPU time is the limiting resource, DE is expected to improve the number of solved instances, assuming the implementation is optimized for speed (e.g. using C++). However, our is not designed to measure this effect, since we implemented in Python, which is typically 100–1,000 times slower than C++, and this low-level bottleneck could hide the effect of speed improvements.

The only meaningful outcome of this experiment is therefore to measure whether DE+PO is better than DE, and if GUCT [*]-Normal2 continues to dominate the other algorithms when DE is used. Table 1 answers both questions positively: DE+PO tends to perform better than DE alone, and the algorithmic efficiency of GUCT [*]-Normal2 is still superior to other algorithms with DE and DE+PO.

An interesting result observed in our experiment is that the results of GUCT [*]-01 with DE, PO, and DE+PO are *still* massively inferior to GBFS. This indicates that the improvement of GUCT [*]-01 + DE+PO observed by Schulte and Keller is purely an artifact of low-level performance and not a fundamental improvement in search efficiency. Indeed, Schulte and Keller (2014) did not analyze node evaluations nor the results of GUCT [*]-01 + PO (they only analyzed DE and DE+PO). Moreover, it means GUCT [*]-01 *requires* DE, an ad hoc and theoretically under-investigated technique, in order to outperform GBFS.

**Solution Quality**   We discuss the quality (here defined as inverse cost) of the solutions returned by each algorithm using the $h^{\text{FF}}$, $h^{\text{add}}$, and $h^{\text{max}}$ heuristics. Fig. 1 shows that GUCT [*]-Normal2 returns consistently longer, thus worse, solutions than GBFS does. In contrast, the solution quality tends to be similar between GBFS and other unsuccessful MCTS algorithms. See appendix Fig. S5-S8 for more plots. As the saying goes, "*haste makes waste*," but in a positive sense: for agile search, we claim that a successful exploration must sacrifice the solution quality for faster search.

While Schulte and Keller (2014) claimed that exploration mechanisms could improve solution quality, this does not necessarily contradict our observations. First, their claim only applies to their evaluation of [G]UCT [*]-01. Our result comparing GUCT [*]-01 and GBFS agrees with their result (Schulte and Keller 2014, Table.2, 143.5 vs 143.57). Second, the IPC score difference in their paper is small ($A^*$:162.81 vs. UCT*:166.8—about 4 instances of best vs worst solution gap) and could result from random tiebreaking.

## 6   Related Work

Due to its focus on adversarial games, MCTS literature typically assumes a bounded reward setting (e.g., 0/1, -1/+1), making applications of UCB1-Normal scarce (e.g., Google Scholar returns 5900 vs. 60 for keyword "UCB1" and "UCB1-Normal", respectively) except a few model-selection applications (McConachie and Berenson 2018). While Gaussian Process MAB (Srinivas et al. 2010) has been used with MCTS for sequential decision making in continuous space search and robotics (Kim et al. 2020), it is significantly different from discrete search spaces like in classical planning. Bayes-UCT2 (Tesauro, Rajan, and Segal 2010) was only evaluated on a synthetic tree and indeed was often outperformed by the base UCT (Imagawa and Kaneko 2016).

MABs may provide a rigorous theoretical tool to analyze the behavior of a variety of existing randomized enhancements for agile/satisficing search that tackle the exploration-exploitation dilemma. $\epsilon$-greedy GBFS was indeed inspired by MABs (Valenzano et al. 2014, Sec. 2). GUCT-Normal2 encourages exploration in nodes further from the goal, which tend to be close to the initial state. This behavior is similar to that of Diverse Best First Search (Imai and Kishimoto 2011), which stochastically enters an "exploration mode" that expands a node with a smaller $g$ value more often. This reverse ordering is unique from other diversified search algorithms, including $\epsilon$-GBFS, Type-GBFS (Xie,

Müller, and Holte 2015), and Softmin-Type-GBFS (Kuroiwa and Beck 2022), which selects $g$ rather uniformly during the exploration.

Theoretical guarantees of MABs require modifications in tree-based algorithms (e.g. MCTS) due to non-i.i.d. sampling from the subtrees (Coquelin and Munos 2007; Munos et al. 2014). Incorporating the methods developed in the MAB community to counter this bias in the subtree samples is an important direction for future work.

MDP and Reinforcement Learning literature often use discounting to avoid the issue of divergent cumulative reward: when the upper bound of step-wise reward is known to be $R$, then the maximum cumulative reward goes to $\infty$ with infinite horizon, while the discounting with $\gamma$ makes it below $\frac{R}{1-\gamma}$, allowing the application of UCB1. Although it addresses the numerical issue and UCB1's theoretical requirement, it no longer optimizes the cumulative objective.

## 7   Conclusion

We examined the theoretical assumptions of existing bandit-based exploration mechanisms for classical planning, and showed that ad hoc design decisions can invalidate theoretical guarantees and harm performance. We presented GUCT-Normal2, a classical planning algorithm combining MCTS and UCB1-Normal2, and analyzed it both theoretically and empirically. The theoretical analysis of its regret bound revealed that, despite its worst-case polynomial bound, in practice it outperforms logarithmically-bounded UCB1-Normal due to the unique aspect of the target application (classical planning). Most importantly, GUCT-Normal2 outperforms GBFS, making it the first bandit-based MCTS to outperform traditional algorithms. Future work includes combinations with other enhancements for agile search including novelty metric (Lipovetzky and Geffner 2017), as well as C++ re-implementation and the comparison with the state-of-the-art.

Our study showcases the importance of considering theoretical assumptions when choosing the correct bandit algorithm for a given application. However, this does not imply that UCB1-Normal is the end of the story: for example, while the Gaussian assumption is *sufficient* for cost-to-go estimates in classical planning, it is not *necessary* for justifying its application to classical planning. The Gaussian assumption implies that rewards can be any value in $[-\infty, \infty]$, which is an under-specification for non-negative cost-to-go estimates. Future work will explore bandits that reflect the assumptions in classical planning with even greater fidelity.

### Acknowledgments

This work was supported through DTIC contract FA8075-18-D-0008, Task Order FA807520F0060, Task 4 - Autonomous Defensive Cyber Operations (DCO) Research & Development (R&D).

### References

Alkhazraji, Y.; Frorath, M.; Grützner, M.; Helmert, M.; Liebetraut, T.; Mattmüller, R.; Ortlieb, M.; Seipp, J.; Springenberg, T.; Stahl, P.; and Wülfing, J. 2020. Pyperplan. `https://`

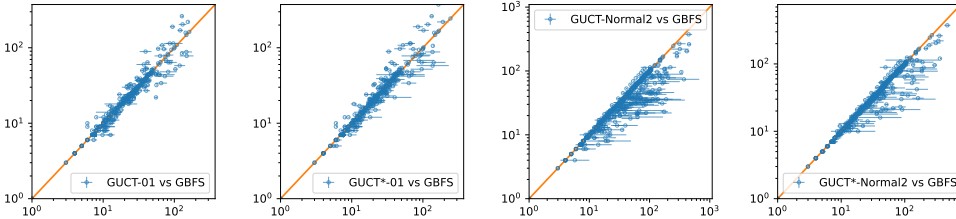

Figure 1: Comparing solution length of GUCT-based algorithms ($x$-axis) against GBFS ($y$-axis) using $h^{\mathrm{FF}}$.

doi.org/10.5281/zenodo.3700819. Accessed: 2022-06-08.

Asai, M.; and Fukunaga, A. 2017. Exploration Among and Within Plateaus in Greedy Best-First Search. In *Proc. of ICAPS*. Pittsburgh, USA.

Audibert, J.-Y.; Munos, R.; and Szepesvári, C. 2009. Exploration–Exploitation Tradeoff using Variance Estimates in Multi-Armed Bandits. *Theoretical Computer Science*, 410(19): 1876–1902.

Auer, P.; Cesa-Bianchi, N.; and Fischer, P. 2002. Finite-Time Analysis of the Multiarmed Bandit Problem. *Machine Learning*, 47(2-3): 235–256.

Bellemare, M. G.; Naddaf, Y.; Veness, J.; and Bowling, M. 2015. The Arcade Learning Environment: An Evaluation Platform for General Agents (Extended Abstract). In Yang, Q.; and Wooldridge, M. J., eds., *Proc. of IJCAI*, 4148–4152. AAAI Press.

Bonet, B.; and Geffner, H. 2001. Planning as Heuristic Search. *Artificial Intelligence*, 129(1): 5–33.

Bush, R. R.; and Mosteller, F. 1953. A Stochastic Model with Applications to Learning. *The Annals of Mathematical Statistics*, 559–585.

Coquelin, P.-A.; and Munos, R. 2007. Bandit Algorithms for Tree Search. In *Proc. of UAI*, 67–74.

Dijkstra, E. W. 1959. A Note on Two Problems in Connexion with Graphs. *Numerische mathematik*, 1(1): 269–271.

Ferber, P.; Cohen, L.; Seipp, J.; and Keller, T. 2022a. Learning and Exploiting Progress States in Greedy Best-First Search. In *Proc. of IJCAI*.

Ferber, P.; Geißer, F.; Trevizan, F.; Helmert, M.; and Hoffmann, J. 2022b. Neural Network Heuristic Functions for Classical Planning: Bootstrapping and Comparison to Other Methods. In *Proc. of ICAPS*.

Ferber, P.; Helmert, M.; and Hoffmann, J. 2020. Neural Network Heuristics for Classical Planning: A Study of Hyperparameter Space. In *Proc. of ECAI*, 2346–2353.

Fikes, R. E.; Hart, P. E.; and Nilsson, N. J. 1972. Learning and Executing Generalized Robot Plans. *Artificial Intelligence*, 3(1-3): 251–288.

Garrett, C. R.; Kaelbling, L. P.; and Lozano-Pérez, T. 2016. Learning to Rank for Synthesizing Planning Heuristics. In *Proc. of IJCAI*, 3089–3095.

Gehring, C.; Asai, M.; Chitnis, R.; Silver, T.; Kaelbling, L. P.; Sohrabi, S.; and Katz, M. 2022. Reinforcement Learning for Classical Planning: Viewing Heuristics as Dense Reward Generators. In *Proc. of ICAPS*.

Gelly, S.; and Silver, D. 2011. Monte-Carlo Tree Search and Rapid Action Value Estimation in Computer Go. *Artificial Intelligence*, 175(11): 1856–1875.

Hart, P. E.; Nilsson, N. J.; and Raphael, B. 1968. A Formal Basis for the Heuristic Determination of Minimum Cost Paths. *Systems Science and Cybernetics, IEEE Transactions on*, 4(2): 100–107.

Helmert, M. 2006. The Fast Downward Planning System. *J. Artif. Intell. Res.(JAIR)*, 26: 191–246.

Heusner, M.; Keller, T.; and Helmert, M. 2017. Understanding the Search Behaviour of Greedy Best-First Search. In *Proc. of SOCS*, volume 8.

Heusner, M.; Keller, T.; and Helmert, M. 2018a. Best-Case and Worst-Case Behavior of Greedy Best-First Search. In *Proc. of IJCAI*.

Heusner, M.; Keller, T.; and Helmert, M. 2018b. Search Progress and Potentially Expanded States in Greedy Best-First Search. In *Proc. of IJCAI*.

Hoffmann, J.; and Nebel, B. 2001. The FF Planning System: Fast Plan Generation through Heuristic Search. *J. Artif. Intell. Res.(JAIR)*, 14: 253–302.

Imagawa, T.; and Kaneko, T. 2016. Monte carlo tree search with robust exploration. In *International Conference on Computers and Games*, 34–46. Springer.

Imai, T.; and Kishimoto, A. 2011. A Novel Technique for Avoiding Plateaus of Greedy Best-First Search in Satisficing Planning. In *Proc. of AAAI*.

Kaufmann, E.; Cappé, O.; and Garivier, A. 2012. On Bayesian Upper Confidence Bounds for Bandit Problems. In *Proc. of AISTATS*, 592–600. PMLR.

Keller, T.; and Helmert, M. 2013. Trial-Based Heuristic Tree Search for Finite Horizon MDPs. In *Proc. of ICAPS*.

Kim, B.; Lee, K.; Lim, S.; Kaelbling, L.; and Lozano-Pérez, T. 2020. Monte Carlo Tree Search in Continuous Spaces using Voronoi Optimistic Optimization with Regret Bounds. In *Proc. of AAAI*, volume 34, 9916–9924.

Kishimoto, A.; Bouneffouf, D.; Marinescu, R.; Ram, P.; Rawat, A.; Wistuba, M.; Palmes, P.; and Botea, A. 2022. Bandit Limited Discrepancy Search and Application to Machine Learning Pipeline Optimization. In *Proc. of AAAI*, volume 36, 10228–10237.

Kishimoto, A.; Zhou, R.; and Imai, T. 2012. Diverse Depth-First Search in Satisificing Planning. In *Proc. of SOCS*, volume 3.

Kocsis, L.; and Szepesvári, C. 2006. Bandit Based Monte-Carlo Planning. In *Proc. of ECML*, 282–293. Springer.

Kuroiwa, R.; and Beck, J. C. 2022. Biased Exploration for Satisificing Heuristic Search. In *Proc. of ICAPS*.

Lai, T. L.; Robbins, H.; et al. 1985. Asymptotically Efficient Adaptive Allocation Rules. *Advances in Applied Mathematics*, 6(1): 4–22.

Lipovetzky, N.; and Geffner, H. 2017. Best-First Width Search: Exploration and Exploitation in Classical Planning . In *Proc. of AAAI*.

Lipovetzky, N.; Ramírez, M.; and Geffner, H. 2015. Classical Planning with Simulators: Results on the Atari Video Games. In *Proc. of IJCAI*.

McConachie, D.; and Berenson, D. 2018. Estimating Model Utility for Deformable Object Manipulation using Multiarmed Bandit Methods. *IEEE Transactions on Automation Science and Engineering*, 15(3): 967–979.

Munos, R.; et al. 2014. From Bandits to Monte-Carlo Tree Search: The Optimistic Principle Applied to Optimization and Planning. *Foundations and Trends® in Machine Learning*, 7(1): 1–129.

Nakhost, H.; and Müller, M. 2009. Monte-Carlo Exploration for Deterministic Planning. In *Proc. of IJCAI*.

Richter, S.; and Helmert, M. 2009. Preferred operators and deferred evaluation in satisficing planning. In *Proc. of ICAPS*, volume 19, 273–280.

Richter, S.; Westphal, M.; and Helmert, M. 2011. LAMA 2008 and 2011. In *Proc. of IPC*, 117–124.

Rivlin, O.; Hazan, T.; and Karpas, E. 2019. Generalized Planning With Deep Reinforcement Learning. In *Proc. of PRL*.

Robbins, H. 1952. Some Aspects of the Sequential Design of Experiments. *Bulletin of the American Mathematical Society*, 58(5): 527–535.

Schulte, T.; and Keller, T. 2014. Balancing Exploration and Exploitation in Classical Planning. In *Proc. of SOCS*.

Shen, W.; Trevizan, F.; and Thiébaux, S. 2020. Learning Domain-Independent Planning Heuristics with Hypergraph Networks. In *Proc. of ICAPS*, volume 30, 574–584.

Silver, D.; Huang, A.; Maddison, C. J.; Guez, A.; Sifre, L.; Van Den Driessche, G.; Schrittwieser, J.; Antonoglou, I.; Panneershelvam, V.; Lanctot, M.; et al. 2016. Mastering the Game of Go with Deep Neural Networks and Tree Search. *Nature*, 529(7587): 484–489.

Srinivas, N.; Krause, A.; Kakade, S. M.; and Seeger, M. W. 2010. Gaussian Process Optimization in the Bandit Setting: No Regret and Experimental Design. In Fürnkranz, J.; and Joachims, T., eds., *Proc. of ICML*, 1015–1022. Omnipress.

Tesauro, G.; Rajan, V.; and Segal, R. 2010. Bayesian Inference in Monte-Carlo Tree Search. In *Proc. of UAI*, 580–588.

Thompson, W. R. 1933. On the Likelihood that One Unknown Probability Exceeds Another in View of the Evidence of Two Samples. *Biometrika*, 25(3-4): 285–294.

Toyer, S.; Trevizan, F.; Thiébaux, S.; and Xie, L. 2018. Action Schema Networks: Generalised Policies with Deep Learning. In *Proc. of AAAI*, volume 32.

Valenzano, R. A.; Schaeffer, J.; Sturtevant, N. R.; and Xie, F. 2014. A Comparison of Knowledge-Based GBFS Enhancements and Knowledge-Free Exploration. In *Proc. of ICAPS*.

Xie, F.; Müller, M.; and Holte, R. C. 2014. Adding Local Exploration to Greedy Best-First Search in Satisficing Planning. In *Proc. of AAAI*, 2388–2394.

Xie, F.; Müller, M.; and Holte, R. C. 2015. Understanding and Improving Local Exploration for GBFS. In *Proc. of ICAPS*, 244–248.

Xie, F.; Müller, M.; Holte, R. C.; and Imai, T. 2014. Type-Based Exploration with Multiple Search Queues for Satisficing Planning. In *Proc. of AAAI*.

Xie, F.; Nakhost, H.; and Müller, M. 2012. Planning Via Random Walk-Driven Local Search. In *Proc. of ICAPS*.

# Appendix

## S1 Domain-Independent Heuristics in Classical Planning

A domain-independent heuristic function $h$ in classical planning is a function of a state $s$ and the problem $[P, A, I, G]$, but the notation $h(s)$ usually omits the latter. In addition to what we discussed in the main article, this section also uses a notation $h(s, G)$. It returns an estimate of the cumulative cost from $s$ to one of the goal states (states that satisfy $G$), typically through a symbolic, non-statistical means including problem relaxation and abstraction. Notable state-of-the-art functions that appear in this paper includes $h^{\mathrm{FF}}, h^{\max}, h^{\mathrm{add}}, h^{\mathrm{GC}}$ (Hoffmann and Nebel 2001; Bonet and Geffner 2001; Fikes, Hart, and Nilsson 1972).

A significant class of heuristics is called delete relaxation heuristics, which solve a relaxed problem which does not contain delete effects, and then returns the cost of the solution of the relaxed problem as an output. The cost of the optimal solution of a delete relaxed planning problem from a state $s$ is denoted by $h^+(s)$, but this is too expensive to compute in practice (NP-complete) (Bylander 1996). Therefore, practical heuristics typically try to obtain its further relaxations that can be computed in polynomial time.

One such admissible heuristic based on delete-relaxation is called $h^{\max}$ (Bonet and Geffner 2001) that is recursively defined as follows:

$$
h^{\max}(s, G) = \max_{p \in G} \begin{cases} 0 \text{ if } p \in s. \text{ Otherwise,} \\ \min_{\{a \in A | p \in \mathrm{ADD}(a)\}} \\ \quad \left[ \mathrm{C}(a) + h^{\mathrm{add}}(s, \mathrm{PRE}(a)) \right]. \end{cases}
\tag{1}
$$

Its inadmissible variant is called additive heuristics $h^{\mathrm{add}}$ (Bonet and Geffner 2001) that is recursively defined as follows:

$$
h^{\mathrm{add}}(s, G) = \sum_{p \in G} \begin{cases} 0 \text{ if } p \in s. \text{ Otherwise,} \\ \min_{\{a \in A | p \in \mathrm{ADD}(a)\}} \\ \quad \left[ \mathrm{C}(a) + h^{\mathrm{add}}(s, \mathrm{PRE}(a)) \right]. \end{cases}
\tag{2}
$$

Another inadmissible delete-relaxation heuristics called $h^{\mathrm{FF}}$ (Hoffmann and Nebel 2001) is defined based on another heuristics $h$, such as $h = h^{\mathrm{add}}$, as a subprocedure. For each unachieved subgoal $p \in G \setminus s$, the action $a$ that adds $p$ with the minimal $[\mathrm{C}(a) + h(s, \mathrm{PRE}(a))]$ is conceptually "the cheapest action that achieves a subgoal $p$ for the first time under delete relaxation", called the *cheapest achiever / best supporter* $\mathrm{bs}(p, s, h)$ of $p$. $h^{\mathrm{FF}}$ is defined as the sum of actions in a relaxed plan $\Pi^+$ constructed as follows:

$$
h^{\mathrm{FF}}(s, G, h) = \sum_{a \in \Pi^+(s, G, h)} \mathrm{C}(a)
\tag{3}
$$

$$
\Pi^+(s, G, h) = \bigcup_{p \in G} \begin{cases} \emptyset \text{ if } p \in s. \text{ Otherwise,} \\ \{a\} \cup \Pi^+(s, \mathrm{PRE}(a)) \\ \quad \text{where } a = \mathrm{bs}(p, s, h). \end{cases}
\tag{4}
$$

$$
\mathrm{bs}(p, s, h) = \underset{\{a \in A | p \in \mathrm{ADD}(a)\}}{\arg \min} \left[ \mathrm{C}(a) + h(s, \mathrm{PRE}(a)) \right].
\tag{5}
$$

Goal Count heuristics $h^{\mathrm{GC}}$ is a simple heuristic proposed in (Fikes, Hart, and Nilsson 1972) that counts the number of propositions that are not satisfied yet. [condition] is a cronecker's delta / indicator function that returns 1 when the condition is satisfied.

$$
h^{\mathrm{GC}}(s, G) = \sum_{p \in G} [\![ p \notin s ]\!].
\tag{6}
$$

## S2 Detailed Explanation for the Base MCTS for Graph Search

Alg. 1 shows the pseudocode of MCTS adjusted for graph search (Schulte and Keller 2014). Aside from what was described from the main section, it has a node-locking mechanism that avoids duplicate search effort.

Following THTS, our MCTS has a hash table that implements a *CLOSE* list and a *Transposition Table* (TT). A CLOSE list stores the generated states and avoids instantiating nodes with duplicate states. A TT stores various information about the states such as the parent information and the action used at the parent. The close list is implemented by a lock mechanism.

Since an efficient graph search algorithm must avoid visiting the same state multiple times, MCTS for graph search marks certain nodes as *locked*, and excludes them from the selection candidates. A node is locked either (1) when a node is a dead-end that will never reach a goal (detected by having no applicable actions, by a heuristic function, or other facilities), (2) when there is a node with the same state in the search tree with a smaller g-value, (3) when all of its children are locked, or (4) when a node is a goal (relevant in an anytime iterated search setting (Richter, Thayer, and Ruml 2010; Richter, Westphal, and Helmert 2011), but not in this paper). Thus, in the expansion step, when a generated node $n$ has the same state as a node $n'$ already in the search tree, MCTS discards $n$ if $g(n) > g(n')$, else moves the subtree of $n'$ to $n$ and marks $n'$ as locked. It also implicitly detects a cycle, as this is identical to the duplicate detection in Dijkstra/A*/GBFS.

The queueing step backpropagates necessary information from the leaf to the root. Efficient backpropagation uses a priority queue ordered by descending g-value. The queue is initialized with the expanded node $p$; each newly generated node $n$ that is not discarded is inserted into the queue, and if a node $n'$ for the same state was already present in the tree it is also inserted into the queue. In each backpropagation iteration, (1) the enqueued node with the highest g-value is popped, (2) its information is updated by aggregating its children's information (including the lock status), (3) and its parent is queued.

## S3 Proof of Bandit Algorithms

To help understand the proof of UCB1-Normal2, we first describe the general procedure for proving the regret of bandit algorithms, demonstrate the proof of UCB1 using this scheme, then finally show the proof of UCB1-Normal2.

The ingredients for proving an upper/lower confidence bound are as follows:

- **Ingredient 1: A specification of reward distributions.** For example, in the standard UCB1 (Auer, Cesa-Bianchi, and Fischer 2002), one assumes a reward distribution bounded in $[0, b]$. Different algorithms assume different reward distributions, and in general, more information about the distribution gives a tighter bound (and faster convergence). For example, one can assume an unbounded distribution with known variance, etc.

- **Ingredient 2: A concentration inequality.** It is also called a tail probability bound. For example, in the standard UCB1, one uses Hoeffding's inequality. Different algorithms use different inequalities to prove the bound. Examples include the Chernoff bound, Chebishev's inequality, Bernstein's inequality, Bennett's inequality, etc. Note that the inequality may be two-sided or one-sided.

The general procedure for proving the bound is as follows.

1. Write down the concentration inequality.
   - $P(|X - \mathbb{E}[X]| \geq \epsilon) \leq F(\epsilon)$. (two-sided)
   - $P(X - \mathbb{E}[X] \geq \epsilon) \leq F(\epsilon)$. (one-sided, upper)
   - $P(\mathbb{E}[X] - X \geq \epsilon) \leq F(\epsilon)$. (one-sided, lower)

   $F$ is an inequality-specific form. This step may be sometimes missing, depending on which inequality you use.

2. Turn the inequality into a version for a sum of independent variables $S_n = \sum_{i=1}^{n} X_i$.

   $$P(|S_n - \mathbb{E}[S_n]| \geq \epsilon) \leq G(\epsilon).$$

   $G$ is an inequality-specific form.

3. Divide the error by $n$ and use $\delta = \frac{\epsilon}{n}$. This makes the statement about the sum $S_n$ into one for the mean $\mu_n = \frac{1}{n} \sum_{i=1}^{n} X_i$. Note that $\mathbb{E}[\mu_n] = \mathbb{E}[X]$ if $X_i$ are i.i.d..

   $$P(|\mu_n - \mathbb{E}[\mu_n]| \geq \frac{\epsilon}{n} = \delta) \leq G(n\delta)$$

4. Simplify the inequality based on the assumptions made in the reward distribution, e.g., bounds, mean, variance.

5. Expand $|\mu_n - \mathbb{E}[\mu_n]| \geq \delta$ into $\delta \geq \mu_n - \mathbb{E}[\mu_n] \geq -\delta$.

6. Change the notations to model the bandit problem because each concentration inequality is a general statement about RVs. Before this step, the notation was:
   - $n$ (number of samples)
   - $\mu_n = \frac{1}{n} \sum_{i=1}^{n} X_i$
   - $\mathbb{E}[\mu_n] = \mathbb{E}[X_1] = \ldots = \mathbb{E}[X_n]$
   - $\frac{1}{n} \sum_{i=1}^{n} (X_i - \mathbb{E}[X_i])^2$
   - $\mathrm{Var}[X_1] = \ldots = \mathrm{Var}[X_n]$

   After the change, they correspond to:
   - $n_i$ (number of pulls of arm $i$).
   - $\hat{\mu}_i$ (sample mean of arm $i$ from $n_i$ pulls),
   - $\mu_i$ (true mean of arm $i$),
   - $\hat{\sigma}_i^2$ (sample variance of arm $i$ from $n_i$ pulls),
   - $\sigma_i^2$ (true variance of arm $i$),

7. Let $i$ be a suboptimal arm, $*$ be an optimal arm, $\mathrm{UCB}_i = \hat{\mu}_i + \delta$, and $\mathrm{LCB}_i = \hat{\mu}_i - \delta$. Derive the relationship between $\delta$ and the gap $\Delta_i = \mu_i - \mu_*$ so that the following conditions for the best arm holds:
   - $\mathrm{UCB}_i \leq \mathrm{UCB}_*$ (for maximization)
   - $\mathrm{LCB}_i \geq \mathrm{LCB}_*$ (for minimization)

   This results in $2\delta \leq \Delta_i$.

8. Replace the $\delta$ with a formula that becomes an exploration term. For example, in UCB1, $\delta = \sqrt{\frac{2 \log T}{n_i}}$.

9. Derive the lower bound $L$ for $n_i$ from $2\delta \leq \Delta_i$.

10. Find the upper-bound of the probability of selecting a sub-optimal arm $i$. This is typically done by a union-bound argument.

11. Derive the upper bound of the expected number of pulls $\mathbb{E}[n_i]$ of a suboptimal arm $i$ using a triple loop summation. This is typically the heaviest part that needs mathematical tricks. The tricks do not seem generally transferable between approaches.

12. Finally, derive an upper bound of the regret $T\mu_* - \sum_{i=1}^{K} \mu_i \mathbb{E}[n_i]$ by

    $$T\mu_* - \sum_{i=1}^{K} \mu_i \mathbb{E}[n_i] = \sum_{i=1}^{K} (\mu_* - \mu_i) \mathbb{E}[n_i]$$
    $$= \sum_{i=1}^{K} \Delta_i \mathbb{E}[n_i].$$

### S3.1 The Proof of UCB1

1. UCB1 uses Hoeffding's inequality, which is already defined for a sum of RVs, thus the first step is skipped.

2. UCB1 assumes a reward distribution with a known bound. According to Hoeffding's inequality, given RVs $X_1 \ldots X_n$, where $X_i \in [l_i, u_i]$, and their sum $S_n = \sum_{i=1}^{n} X_i$,

   $$P(S_n - \mathbb{E}[S_n] \geq \epsilon) \leq \exp - \frac{2\epsilon^2}{\sum_{i=1}^{n} (u_i - l_i)^2}.$$
   $$P(\mathbb{E}[S_n] - S_n \geq \epsilon) \leq \exp - \frac{2\epsilon^2}{\sum_{i=1}^{n} (u_i - l_i)^2}.$$

   We focus on $P(S_n - \mathbb{E}[S_n] \geq \epsilon)$ to avoid repetition.

3. Using $\delta = \frac{\epsilon}{n}$ and $\mu_n = \frac{S_n}{n}$,

   $$P(\mu_n - \mathbb{E}[\mu_n] \geq \delta) \leq \exp - \frac{2n^2\delta^2}{\sum_{i=1}^{n} (u_i - l_i)^2}.$$

4. UCB1 assumes $X_i$ are i.i.d. copies, thus $\forall i; u_i - l_i = c$.

   $$P(\mu_n - \mathbb{E}[\mu_n] \geq \delta) \leq \exp - \frac{2n^2\delta^2}{nc^2} = \exp - \frac{2n\delta^2}{c^2}.$$

5. Expanding the two-sided error:

   $$\delta \geq \mu_n - \mathbb{E}[\mu_n] \geq -\delta.$$

6. Changing the notation:

   $$\delta \geq \hat{\mu}_i - \mu_i \geq \delta.$$

7. Adding $\mu_i - \delta$ to both sides,

$$\mu_i \geq \hat{\mu}_i - \delta = \text{LCB}_i(T, n_i) \geq \mu_i - 2\delta.$$

Substituting $i = *$ (optimal arm), the first inequality is

$$\mu_* \geq \hat{\mu}_* - \delta = \text{LCB}_*(T, n_*).$$

**Assuming** $2\delta \leq \Delta_i = \mu_i - \mu_*$, the second inequality is

$$\text{LCB}_i(T, n_i) \geq \mu_i - 2\delta \geq \mu_i - \Delta_i = \mu_*.$$

Therefore

$$\text{LCB}_i(T, n_i) \geq \mu_* \geq \text{LCB}_*(T, n_*).$$

8. Let $\delta = c\sqrt{\frac{2\log T}{n_i}}$. Then

$$P(\mu_{n_i} - \mathbb{E}[\mu_{n_i}] \geq \delta) \leq \exp -\frac{2nc^2 \frac{2\log T}{n_i}}{c^2} = T^{-4}.$$

9. From $2\delta \leq \Delta_i$, considering $n_i$ is an integer,

$$2c\sqrt{\frac{2\log T}{n_i}} \leq \Delta_i \Leftrightarrow 4c^2 \frac{2\log T}{n_i} \leq \Delta_i^2$$

$$\Leftrightarrow \frac{8c^2 \log T}{\Delta_i^2} \leq \left\lceil \frac{8c^2 \log T}{\Delta_i^2} \right\rceil = L \leq n_i.$$

10. $\text{LCB}_i(T, n_i) \geq \mu_* \geq \text{LCB}_*(T, n_*)$ does not hold when either inequality does not hold. $\text{LCB}_i(T, n_i) \geq \mu_*$ does not hold with probability less than $T^{-4}$. $\mu_* \geq \text{LCB}_i(T, n_*)$ does not hold with probability less than $T^{-4}$. Thus, by union-bound (probability of disjunctions),

$$P(\text{LCB}_i(T, n_i) \leq \text{LCB}_*(T, n_*)) \leq 2T^{-4}.$$

11. Assume we followed the UCB1 strategy, i.e., we pulled the arm that minimizes the LCB. The expected number of pulls $\mathbb{E}[n_i]$ from a suboptimal arm $i$ is as follows. Note that for $K$ arms, every arm is at least pulled once.

$$\mathbb{E}[n_i] = 1 + \sum_{t=K+1}^{T} P(i \text{ is pulled at time } t)$$

$$\leq L + \sum_{t=K+1}^{T} P(i \text{ is pulled at time } t \wedge n_i > L)$$

$$= L + \sum_{t=K+1}^{T} P(\forall j; \text{LCB}_j(t, n_j) \geq \text{LCB}_i(t, n_i))$$

$$\leq L + \sum_{t=K+1}^{T} P(\text{LCB}_*(t, n_*) \geq \text{LCB}_i(t, n_i))$$

$$\leq L + \sum_{t=K+1}^{T} P(\exists u, v; \text{LCB}_*(t, u) \geq \text{LCB}_i(t, v))$$

$$\leq L + \sum_{t=K+1}^{T} \sum_{u=1}^{t-1} \sum_{v=L}^{t-1} P(\text{LCB}_*(t, u) \geq \text{LCB}_i(t, v))$$

$$\leq L + \sum_{t=K+1}^{T} \sum_{u=1}^{t-1} \sum_{v=L}^{t-1} 2t^{-4}$$

$$\leq L + \sum_{t=1}^{\infty} \sum_{u=1}^{t} \sum_{v=1}^{t} 2t^{-4} = L + \sum_{t=1}^{\infty} t^2 \cdot 2t^{-4}$$

$$= L + 2\sum_{t=1}^{\infty} t^{-2} = L + 2 \cdot \frac{\pi}{6} = L + \frac{\pi}{3}$$

$$\leq c^2 \frac{8\log T}{\Delta_i^2} + 1 + \frac{\pi}{3} \quad \because \lceil x \rceil \leq x + 1$$

12. The regret is

$$T\mu_* - \sum_{i=1}^{K} \mu_i \mathbb{E}[n_i] = \sum_{i=1}^{K}(\mu_* - \mu_i)\mathbb{E}[n_i] = \sum_{i=1}^{K} \Delta_i \mathbb{E}[n_i]$$

$$\leq \sum_{i=1}^{K} \Delta_i \left( c^2 \frac{8\log T}{\Delta_i^2} + 1 + \frac{\pi}{3} \right)$$

$$\leq \sum_{i=1}^{K} \left( c^2 \frac{8\log T}{\Delta_i} + \left(1 + \frac{\pi}{3}\right)\Delta_i \right).$$

## S3.2 Preliminary for the Proof of UCB1-Normal2

Our analysis begins with a definition of Sub-Gaussian distributions.

**Definition 1.** *(Vershynin 2018, Proposition 2.5.2, (iv)) A distribution $p(x)$ is* sub-Gaussian *when*

$$\exists t > 0; \mathbb{E}[\exp x^2/t^2] < 2.$$

**Theorem 1.** *A Gaussian distribution with 0-mean $\mathcal{N}(0, \sigma^2)$ (without loss of generality) is sub-Gaussian.*

*Proof.*

$$p(x) = \mathcal{N}(0, \sigma^2) = \frac{1}{\sqrt{2\pi\sigma^2}} \exp -\frac{x^2}{2\sigma^2}.$$

$$\mathbb{E}[\exp x^2/t^2] = \int_{\mathbb{R}} \exp \frac{x^2}{t^2} \frac{1}{\sqrt{2\pi\sigma^2}} \exp -\frac{x^2}{2\sigma^2} dx$$

$$= \frac{1}{\sqrt{2\pi\sigma^2}} \int_{\mathbb{R}} \exp -x^2 \left( \frac{1}{2\sigma^2} - \frac{1}{t^2} \right) dx$$

$$= \frac{1}{\sqrt{2\pi\sigma^2}} \int_{\mathbb{R}} \exp -\frac{x^2}{C^2} dx$$

$$= \frac{1}{\sqrt{2\pi\sigma^2}} \int_{\mathbb{R}} \exp -y^2 C dy \quad \left( \frac{x}{C} = y \Leftrightarrow dx = C dy \right)$$

$$= \frac{C}{\sqrt{2\pi\sigma^2}} \sqrt{\pi}$$

$$= \frac{C}{\sqrt{2\sigma^2}}.$$

Where

$$\frac{1}{C^2} = \frac{1}{2\sigma^2} - \frac{1}{t^2}$$

$$\Leftrightarrow C^2 = \frac{2\sigma^2 t^2}{t^2 - 2\sigma^2}.$$

To show $\mathbb{E}[\exp x^2/t^2] < 2$,

$$\mathbb{E}[\exp x^2/t^2] = \frac{C}{\sqrt{2\sigma^2}} = \sqrt{\frac{t^2}{t^2 - 2\sigma^2}} < 2,$$
$$\Leftrightarrow t^2 < 4(t^2 - 2\sigma^2),$$
$$\Leftrightarrow \frac{8}{3}\sigma^2 < t^2.$$

$\square$

**Definition 2.** *For a sub-Gaussian RV $x$,*

$$||x|| = \inf\left\{t > 0 \mid \mathbb{E}[\exp x^2/t^2] < 2\right\}.$$

**Corollary 1.** *For $p(x) = \mathcal{N}(0, \sigma^2)$, $||x|| = \sqrt{\frac{8}{3}}\sigma$.*

Next, we review the general Hoeffding's inequality for sub-Gaussian distributions ().

**Theorem 2.** *For independent sub-Gaussian RVs $x_1, \ldots, x_n$, let their sum be $S_n = \sum_{i=1}^{n} x_i$. Then, for any $\epsilon > 0$,*

$$\Pr(|S_n - \mathbb{E}[S_n]| \le \epsilon) \ge 2\exp -\frac{\epsilon^2}{\sum_{i=1}^{n} ||x_i||^2},$$

$$\Pr(S_n - \mathbb{E}[S_n] \le \epsilon) \ge \exp -\frac{\epsilon^2}{\sum_{i=1}^{n} ||x_i||^2},$$

$$\Pr(\mathbb{E}[S_n] - S_n \le \epsilon) \ge \exp -\frac{\epsilon^2}{\sum_{i=1}^{n} ||x_i||^2}.$$

*(Two-sided bounds and one-sided upper/lower bounds, respectively.)*

### S3.3   The Proof of UCB1-Normal2

1. Same as UCB1.
2. According to Hoeffding's inequality for sub-Gaussian RVs $X_1 \ldots X_n$ and their sum $S_n = \sum_{i=1}^{n} X_i$,

$$P(S_n - \mathbb{E}[S_n] \ge \epsilon) \le \exp -\frac{\epsilon^2}{\sum_{i=1}^{n} ||X_i||^2}.$$

3. Using $\delta = \frac{\epsilon}{n}$,

$$P(\mu_n - \mathbb{E}[\mu_n] \ge \delta) \le \exp -\frac{n^2\delta^2}{\sum_{i=1}^{n} ||X_i||^2}.$$

4. We assume $X_i = \mathcal{N}(\mu, \sigma^2)$, thus $||X_i||^2 = \frac{8}{3}\sigma^2$.

$$P(\mu_n - \mathbb{E}[\mu_n] \ge \delta) \le \exp -\frac{3n^2\delta^2}{8n\sigma^2} = \exp -\frac{3n\delta^2}{8\sigma^2}.$$

5. Same as UCB1.
6. Same as UCB1.
7. Same as UCB1.
8. Let $\delta = \hat{\sigma}\sqrt{\log T}$. Then

$$P(A : \mu_{n_i} - \mathbb{E}[\mu_{n_i}] \ge \delta) \le \exp -\frac{3n_i\hat{\sigma}^2 \log T}{8\sigma^2}$$
$$= T^{-\frac{3n_i\hat{\sigma}^2}{8\sigma^2}}.$$

The trick starts here. The formula above is problematic because we do not know the true variance $\sigma^2$. However, if event $B : \frac{n_i\hat{\sigma}^2}{\sigma^2} \ge X$ holds for some $X > 0$, we have

$$T^{-\frac{3n_i\hat{\sigma}^2}{8\sigma^2}} \le T^{-\frac{3}{8}X}.$$

One issue with this approach is that the two events $A, B$ may be correlated. To address the issue, we further upper-bound the probability by union-bound. Let $P(B) = \alpha$ which is close to 1. Then

$$P(\neg(A \wedge B)) = P(\neg A \vee \neg B) \le P(\neg A) + P(\neg B).$$
$$1 - P(A \wedge B) \le 1 - P(A) + P(\neg B).$$
$$P(A) \le P(A \wedge B) + P(\neg B).$$
$$\therefore P(\mu_{n_i} - \mathbb{E}[\mu_{n_i}] \ge \delta) \le T^{-\frac{3}{8}X} + 1 - \alpha.$$

We next obtain $X$ that satisfies $P(B) = \alpha$. We use the fact that $\frac{n_i\hat{\sigma}^2}{\sigma^2}$ follows a Chi-Squared distribution $\chi^2(n_i)$ with a degree of freedom $n_i$. Then $X = \chi^2_{1-\alpha,n_i}$, the upper-tail critical value of $\chi^2$ distribution with degree of freedom $n_i$ and significance level $\alpha$, because

$$P(\neg B) = P(\frac{n_i\hat{\sigma}^2}{\sigma^2} < \chi^2_{1-\alpha,n_i})$$
$$= \chi^2(\frac{n_i\hat{\sigma}^2}{\sigma^2} < \chi^2_{1-\alpha,n_i} \mid n_i) = 1 - \alpha.$$

9. From $2\delta \le \Delta_i$, assuming $n_i$ is an integer and $n_i \ge 2$,

$$\Delta_i^2 \ge 2\hat{\sigma}^2 \log T = \frac{2n_i\hat{\sigma}^2 \log T}{n_i} \ge \frac{2\sigma^2\chi^2_{1-\alpha,n_i} \log T}{n_i}$$
$$\ge \frac{2\sigma^2\chi^2_{1-\alpha,2} \log T}{n_i} = \frac{-4\sigma^2 \log\alpha \log T}{n_i}.$$
$$\therefore n_i \ge \left\lceil \frac{-4\sigma^2 \log\alpha \log T}{\Delta_i^2} \right\rceil = L \ge \frac{-4\sigma^2 \log\alpha \log T}{\Delta_i^2}.$$

Note that we used the fact that $\chi^2_{1-\alpha,n}$ is monotonically increasing for $n$, therefore $\chi^2_{1-\alpha,n} \ge \chi^2_{1-\alpha,2}$ ($n_i \ge 2$), and that $\chi^2_{1-\alpha,2} = -2\log\alpha$:

$$1 - \alpha = \chi^2(X < \chi^2_{1-\alpha,n} \mid n = 2)$$
$$= \frac{\gamma(\frac{2}{2}, \frac{\chi^2_{1-\alpha,2}}{2})}{\Gamma(\frac{2}{2})} = 1 - e^{-\frac{\chi^2_{1-\alpha,2}}{2}}.$$

where $\gamma$ and $\Gamma$ are (incomplete) Gamma functions.

10. Using the same union-bound argument used in UCB1,

$$P(\text{LCB}_i(T, n_i) \le \text{LCB}_*(T, n_*))$$
$$\le 2(T^{-\chi^2_{1-\alpha,n_i}} + 1 - \alpha).$$

11. Assume we followed the UCB1-Normal2 strategy. We use the same argument as UCB1. Assume we pull each arm at least $M$ times in the beginning and $M \le L$.

$$\mathbb{E}[n_i] \le L + \sum_{t=K+1}^{T} P(\exists u, v; \text{LCB}_*(t, u) \ge \text{LCB}_i(t, v))$$

$$\leq L + \sum_{t=K+1}^{T} \sum_{u=1}^{t-1} \sum_{v=L}^{t-1} 2(t^{-\frac{3}{8}\chi^2_{1-\alpha,v}} + 1 - \alpha)$$

$$\leq L + \sum_{t=K+1}^{T} \sum_{u=1}^{t-1} \sum_{v=L}^{t-1} 2(t^{-\frac{3}{8}\chi^2_{1-\alpha,M}} + 1 - \alpha)$$

$$\leq L + \sum_{t=K+1}^{T} \sum_{u=1}^{t} \sum_{v=1}^{t} 2(t^{-\frac{3}{8}\chi^2_{1-\alpha,M}} + 1 - \alpha)$$

$$= L + \sum_{t=K+1}^{T} 2(t^{2-\frac{3}{8}\chi^2_{1-\alpha,M}} + (1 - \alpha)t^2)$$

$$\leq L + 2\sum_{t=1}^{\infty} t^{2-\frac{3}{8}\chi^2_{1-\alpha,M}} + 2(1 - \alpha)\sum_{t=1}^{T} t^2$$

$$= L + 2C + 2(1 - \alpha)\frac{T(T+1)(2T+1)}{6}$$

$$\leq \frac{-4\sigma^2 \log\alpha \log T}{\Delta_i^2} + 1 \quad (\because \lceil x \rceil \leq x + 1)$$

$$+ 2C + \frac{(1-\alpha)T(T+1)(2T+1)}{3}.$$

$C$ is a convergent series when

$$2 - \frac{3}{8}\chi^2_{1-\alpha,M} < -1 \iff 8 < \chi^2_{1-\alpha,M}.$$

You can look up the value of $M$ that guarantees this condition from a numerically computed, so-called $\chi^2$-*table* (Table S1). For example, with $\alpha = 0.99$, $8 < \chi^2_{0.01,M}$, thus $M \geq 2$, and with $\alpha = 0.9$, $8 < \chi^2_{0.1,M}$, thus $M \geq 5$. However, the value of $\alpha$ depends on the problem and is unknown prior to solving the problem.

12. Omitted.

## S4 Statistics after Merging Datasets

Backpropagation in MCTS requires computing the statistics of the samples in the leaf nodes in a subtree of a parent node. To avoid iterating over all leaves of each parent, Backpropagation typically propagates the statistics from the immediate children. This can be seen as merging multiple datasets and compute the statistics of the merged dataset from the statistics of multiple datasets.

In variance-based MCTS algorithms, both the mean and variance are backpropagated. Given two sets of samples $X_1, X_2$, each with an empirical mean $\mu_i$ and $n_i$ elements ($i \in \{1, 2\}$), the empirical mean $\mu_{12}$ of $X_1 \cup X_2$ is given by

$$\mu_{12} = \frac{\sum_{x\in X_1} x + \sum_{x\in X_2} x}{n_1 + n_2} = \frac{n_1\mu_1 + n_2\mu_2}{n_1 + n_2}$$

We obtain NECs by iterating this process over a node's children, although there is a more efficient, incremental method for backpropagating a change in a single child (see appendix). For the variance, we similarly merge the samples. Given individual variances $\sigma_1^2$ and $\sigma_2^2$, the variance $\sigma_{12}^2$ of $X_1 \cup X_2$ (proof available in appendix) is:

$$\sigma_{12}^2 = \frac{n_1\sigma_1^2 + n_2\sigma_2^2 + \frac{n_1 n_2}{n_1+n_2}(\mu_2 - \mu_1)^2}{n_1 + n_2}.$$

Below, we show the formulae and the proofs for this method.

**Theorem 3** (The empirical mean of merged datasets). *Given two sets of samples $X_1, X_2$, each with an empirical mean $\mu_i$ and $n_i$ elements ($i \in \{1, 2\}$), the empirical mean $\mu_{12}$ of $X_1 \cup X_2$ is given by*

$$\mu_{12} = \frac{n_1\mu_1 + n_2\mu_2}{n_1 + n_2}.$$

$$\text{Also,} \quad \mu_{12} = \mu_1 + \frac{n_2}{n_1 + n_2}(\mu_2 - \mu_1).$$

*Proof.*

$$\mu_{12} = \frac{\sum_{x\in X_1} x + \sum_{x\in X_2} x}{n_1 + n_2} = \frac{n_1\mu_1 + n_2\mu_2}{n_1 + n_2}.$$

$\square$

**Theorem 4** (The empirical variance of merged datasets). *Given two sets of samples $X_1, X_2$, each with an empirical mean $\mu_i$, variance $\sigma_i^2$, and $n_i$ elements ($i \in \{1, 2\}$), and $n_{ij} = n_i + n_j$, the empirical variance $\mu_{12}$ of $X_1 \cup X_2$ is given by*

$$\sigma_{12}^2 = \frac{n_1\sigma_1^2 + n_2\sigma_2^2 + \frac{n_1 n_2}{n_1+n_2}(\mu_2 - \mu_1)^2}{n_1 + n_2}.$$

*Proof.*

$$\sigma_{12}^2 = \frac{\sum_{x\in X_1}(x - \mu_{12})^2 + \sum_{x\in X_2}(x - \mu_{12})^2}{n_1 + n_2}.$$

$$\sum_{x\in X_1}(x - \mu_{12})^2$$

$$= \sum_{x\in X_1}\left(x - \left(\mu_1 + \frac{n_2}{n_1+n_2}(\mu_2 - \mu_1)\right)\right)^2$$

$$= \sum_{x\in X_1}\left((x - \mu_1) - \frac{n_2}{n_1+n_2}(\mu_2 - \mu_1)\right)^2$$

$$= \sum_{x\in X_1}(x - \mu_1)^2 - 2\sum_{x\in X_1}(x - \mu_1)\frac{n_2}{n_1+n_2}(\mu_2 - \mu_1)$$

$$+ \sum_{x\in X_1}\left(\frac{n_2}{n_1+n_2}(\mu_2 - \mu_1)\right)^2$$

$$= n_1\sigma_1^2 - 2\cdot 0 + n_1\left(\frac{n_2}{n_1+n_2}(\mu_2 - \mu_1)\right)^2.$$

$$\therefore (n_1 + n_2)\sigma_{12}^2$$

$$= n_1\sigma_1^2 + n_1\left(\frac{n_2}{n_1+n_2}(\mu_2 - \mu_1)\right)^2$$

$$+ n_2\sigma_2^2 + n_2\left(\frac{n_1}{n_1+n_2}(\mu_1 - \mu_2)\right)^2$$

$$= n_1\sigma_1^2 + n_2\sigma_2^2 + \frac{n_1 n_2^2 + n_2 n_1^2}{(n_1+n_2)^2}(\mu_2 - \mu_1)^2$$

$$= n_1\sigma_1^2 + n_2\sigma_2^2 + \frac{n_1 n_2}{n_1+n_2}(\mu_2 - \mu_1)^2.$$

$\square$

| $\alpha$ / $n$ | 0.995 | 0.99 | 0.975 | 0.95 | 0.90 | 0.10 | 0.05 | 0.025 | 0.01 | 0.005 |
|---|---|---|---|---|---|---|---|---|---|---|
| 1 | — | — | 0.001 | 0.004 | 0.016 | 2.706 | 3.841 | 5.024 | 6.635 | 7.879 |
| 2 | 0.010 | 0.020 | 0.051 | 0.103 | 0.211 | 4.605 | 5.991 | 7.378 | 9.210 | 10.597 |
| 3 | 0.072 | 0.115 | 0.216 | 0.352 | 0.584 | 6.251 | 7.815 | 9.348 | 11.345 | 12.838 |
| 4 | 0.207 | 0.297 | 0.484 | 0.711 | 1.064 | 7.779 | 9.488 | 11.143 | 13.277 | 14.860 |
| 5 | 0.412 | 0.554 | 0.831 | 1.145 | 1.610 | 9.236 | 11.070 | 12.833 | 15.086 | 16.750 |
| 6 | 0.676 | 0.872 | 1.237 | 1.635 | 2.204 | 10.645 | 12.592 | 14.449 | 16.812 | 18.548 |
| 7 | 0.989 | 1.239 | 1.690 | 2.167 | 2.833 | 12.017 | 14.067 | 16.013 | 18.475 | 20.278 |
| 8 | 1.344 | 1.646 | 2.180 | 2.733 | 3.490 | 13.362 | 15.507 | 17.535 | 20.090 | 21.955 |
| 9 | 1.735 | 2.088 | 2.700 | 3.325 | 4.168 | 14.684 | 16.919 | 19.023 | 21.666 | 23.589 |
| 10 | 2.156 | 2.558 | 3.247 | 3.940 | 4.865 | 15.987 | 18.307 | 20.483 | 23.209 | 25.188 |
| 11 | 2.603 | 3.053 | 3.816 | 4.575 | 5.578 | 17.275 | 19.675 | 21.920 | 24.725 | 26.757 |
| 12 | 3.074 | 3.571 | 4.404 | 5.226 | 6.304 | 18.549 | 21.026 | 23.337 | 26.217 | 28.300 |
| 13 | 3.565 | 4.107 | 5.009 | 5.892 | 7.042 | 19.812 | 22.362 | 24.736 | 27.688 | 29.819 |
| 14 | 4.075 | 4.660 | 5.629 | 6.571 | 7.790 | 21.064 | 23.685 | 26.119 | 29.141 | 31.319 |
| 15 | 4.601 | 5.229 | 6.262 | 7.261 | 8.547 | 22.307 | 24.996 | 27.488 | 30.578 | 32.801 |
| 16 | 5.142 | 5.812 | 6.908 | 7.962 | 9.312 | 23.542 | 26.296 | 28.845 | 32.000 | 34.267 |
| 17 | 5.697 | 6.408 | 7.564 | 8.672 | 10.085 | 24.769 | 27.587 | 30.191 | 33.409 | 35.718 |
| 18 | 6.265 | 7.015 | 8.231 | 9.390 | 10.865 | 25.989 | 28.869 | 31.526 | 34.805 | 37.156 |
| 19 | 6.844 | 7.633 | 8.907 | 10.117 | 11.651 | 27.204 | 30.144 | 32.852 | 36.191 | 38.582 |
| 20 | 7.434 | 8.260 | 9.591 | 10.851 | 12.443 | 28.412 | 31.410 | 34.170 | 37.566 | 39.997 |
| 21 | 8.034 | 8.897 | 10.283 | 11.591 | 13.240 | 29.615 | 32.671 | 35.479 | 38.932 | 41.401 |
| 22 | 8.643 | 9.542 | 10.982 | 12.338 | 14.041 | 30.813 | 33.924 | 36.781 | 40.289 | 42.796 |
| 23 | 9.260 | 10.196 | 11.689 | 13.091 | 14.848 | 32.007 | 35.172 | 38.076 | 41.638 | 44.181 |
| 24 | 9.886 | 10.856 | 12.401 | 13.848 | 15.659 | 33.196 | 36.415 | 39.364 | 42.980 | 45.559 |
| 25 | 10.520 | 11.524 | 13.120 | 14.611 | 16.473 | 34.382 | 37.652 | 40.646 | 44.314 | 46.928 |
| 26 | 11.160 | 12.198 | 13.844 | 15.379 | 17.292 | 35.563 | 38.885 | 41.923 | 45.642 | 48.290 |
| 27 | 11.808 | 12.879 | 14.573 | 16.151 | 18.114 | 36.741 | 40.113 | 43.195 | 46.963 | 49.645 |
| 28 | 12.461 | 13.565 | 15.308 | 16.928 | 18.939 | 37.916 | 41.337 | 44.461 | 48.278 | 50.993 |
| 29 | 13.121 | 14.256 | 16.047 | 17.708 | 19.768 | 39.087 | 42.557 | 45.722 | 49.588 | 52.336 |
| 30 | 13.787 | 14.953 | 16.791 | 18.493 | 20.599 | 40.256 | 43.773 | 46.979 | 50.892 | 53.672 |
| 40 | 20.707 | 22.164 | 24.433 | 26.509 | 29.051 | 51.805 | 55.758 | 59.342 | 63.691 | 66.766 |
| 50 | 27.991 | 29.707 | 32.357 | 34.764 | 37.689 | 63.167 | 67.505 | 71.420 | 76.154 | 79.490 |
| 60 | 35.534 | 37.485 | 40.482 | 43.188 | 46.459 | 74.397 | 79.082 | 83.298 | 88.379 | 91.952 |
| 70 | 43.275 | 45.442 | 48.758 | 51.739 | 55.329 | 85.527 | 90.531 | 95.023 | 100.425 | 104.215 |
| 80 | 51.172 | 53.540 | 57.153 | 60.391 | 64.278 | 96.578 | 101.879 | 106.629 | 112.329 | 116.321 |
| 90 | 59.196 | 61.754 | 65.647 | 69.126 | 73.291 | 107.565 | 113.145 | 118.136 | 124.116 | 128.299 |
| 100 | 67.328 | 70.065 | 74.222 | 77.929 | 82.358 | 118.498 | 124.342 | 129.561 | 135.807 | 140.169 |

Table S1: $\chi^2_{\alpha,n}$ table.

## S5 Statistics after Retracting a Dataset

In the backpropagation step of MCTS, typically, only a few children of an intermediate node update their statistics (most often a single children). To compute the updated statistics efficiently, we could compute them by retracting the old data of the child(ren) from the merged data and merging the new data for the child(ren), rather than iterating over the children to merge everything from scratch. This can impact the performance when the number of children / the branching factor is high.

**Theorem 5** (The empirical mean after retracting a dataset)**.** *Assume samples $X_1, X_2$ with empirical means $\mu_i$ and number of elements $n_i$ ($i \in \{1, 2\}$). Let their union be $X_{12} = X_1 \cup X_2$, its empirical means $\mu_{12}$, and its number of elements $n_{12} = n_1 + n_2$. $\mu_1$ is given by*

$$\mu_1 = \frac{n_{12}\mu_{12} - n_2\mu_2}{n_{12} - n_2}.$$

**Theorem 6** (The empirical variance after retracting a dataset)**.** *Assume samples $X_1, X_2$ with empirical means $\mu_i$, empirical variance $\sigma_i^2$, and number of elements $n_i$ ($i \in \{1, 2\}$). Let their union be $X_{12} = X_1 \cup X_2$, its empirical mean $\mu_{12}$, its empirical variance $\sigma_{12}^2$, and its number of elements $n_{12} = n_1 + n_2$. $\sigma_1^2$ is given by $\mu_{12}$, $\mu_2$, $\mu_1$, $n_1$, $n_2$, $n_{12}$, $\sigma_{12}^2$, and $\sigma_2^2$ as follows.*

$$\sigma_1^2 = \frac{1}{n_1}\left(n_{12}\sigma_{12}^2 - n_2\sigma_2^2 - \frac{n_2 n_{12}}{n_1}(\mu_{12} - \mu_2)^2\right)$$

*Proof.*

$$n_{12}\sigma_{12}^2 = n_1\sigma_1^2 + n_2\sigma_2^2 + \frac{n_1 n_2}{n_{12}}(\mu_2 - \mu_1)^2.$$

$$\therefore n_1\sigma_1^2 = n_{12}\sigma_{12}^2 - n_2\sigma_2^2 - \frac{n_1 n_2}{n_{12}}(\mu_2 - \mu_1)^2.$$

$$n_1\mu_1 = n_{12}\mu_{12} - n_2\mu_2.$$

$$\therefore \mu_1 - \mu_2 = \frac{n_{12}\mu_{12} - n_2\mu_2}{n_1} - \mu_2$$

$$= \frac{n_{12}\mu_{12} - n_2\mu_2 - n_1\mu_2}{n_1}$$

$$= \frac{n_{12}\mu_{12} - n_2\mu_2 - (n_{12} - n_2)\mu_2}{n_1}$$

$$= \frac{n_{12}}{n_1}(\mu_{12} - \mu_2).$$

$$\therefore n_1\sigma_1^2 = n_{12}\sigma_{12}^2 - n_2\sigma_2^2 - \frac{n_1 n_2}{n_{12}}\frac{n_{12}^2}{n_1^2}(\mu_{12} - \mu_2)^2$$

$$= n_{12}\sigma_{12}^2 - n_2\sigma_2^2 - \frac{n_2 n_{12}}{n_1}(\mu_{12} - \mu_2)^2.$$

$$\square$$

## S6 Further Results

### S6.1 Cumulative Histograms for All Heuristics and All Search Statistics

Fig. S1-S3 shows the cumulative histogram of the number of instances solved by a particular evaluation/expansion/runtime. Although the graphs for the expansion and the runtime are not entirely informative since it is confounded by the node evaluation limit, the general trend is the same between algorithms.

### S6.2 Deferred Heuristic Evaluation

Fig. S4 shows the cumulative histogram of the number of instances solved under a particular evaluation/expansion/runtime by $h^{\mathrm{FF}}$ with/without DE, with/without PO. Although the graphs for the expansion and the runtime are not entirely informative since it is confounded by the node evaluation limit, the general trend is the same between algorithms.

### S6.3 Solution Quality

Fig. S5-S8 shows the complete plot for the solution quality.

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

**Algorithm 1** High-level general MCTS. **Input**: Root node $r$, successor function $S$, NEC $f$, heuristic function $h$, priority queue $Q$ sorted by $g$. Initialize $\forall n; g(n) \leftarrow \infty$.

---

**while** True **do**
    Parent $p \leftarrow r$
    **while** not leaf $p$ **do**    *# Selection*
      $p \leftarrow \arg\min_{n \in S(p)} f(n)$
    $Q \leftarrow \{p\}$
    **for** $n \in S(p)$ **do**    *# Expansion*
      **return** $n$ **if** $n$ is goal.    *# Early goal detection*
      **if** $\exists n'$ already in tree with same state $s_{n'} = s_n$ **then**
        **if** $g(n) > g(n')$ **then**
          **continue**
        Lock $n'$, $S(n) \leftarrow S(n')$, $Q \leftarrow Q \cup \{n, n'\}$
      **else**
        Compute $h(s_n)$    *# Evaluation*
        $Q \leftarrow Q \cup \{n\}$
    **while** $n \leftarrow Q.\text{POPMAX}()$ **do**    *# Backpropagation*
      Update $n$'s statistics and lock status
      $Q \leftarrow Q \cup \{n\text{'s parent}\}$

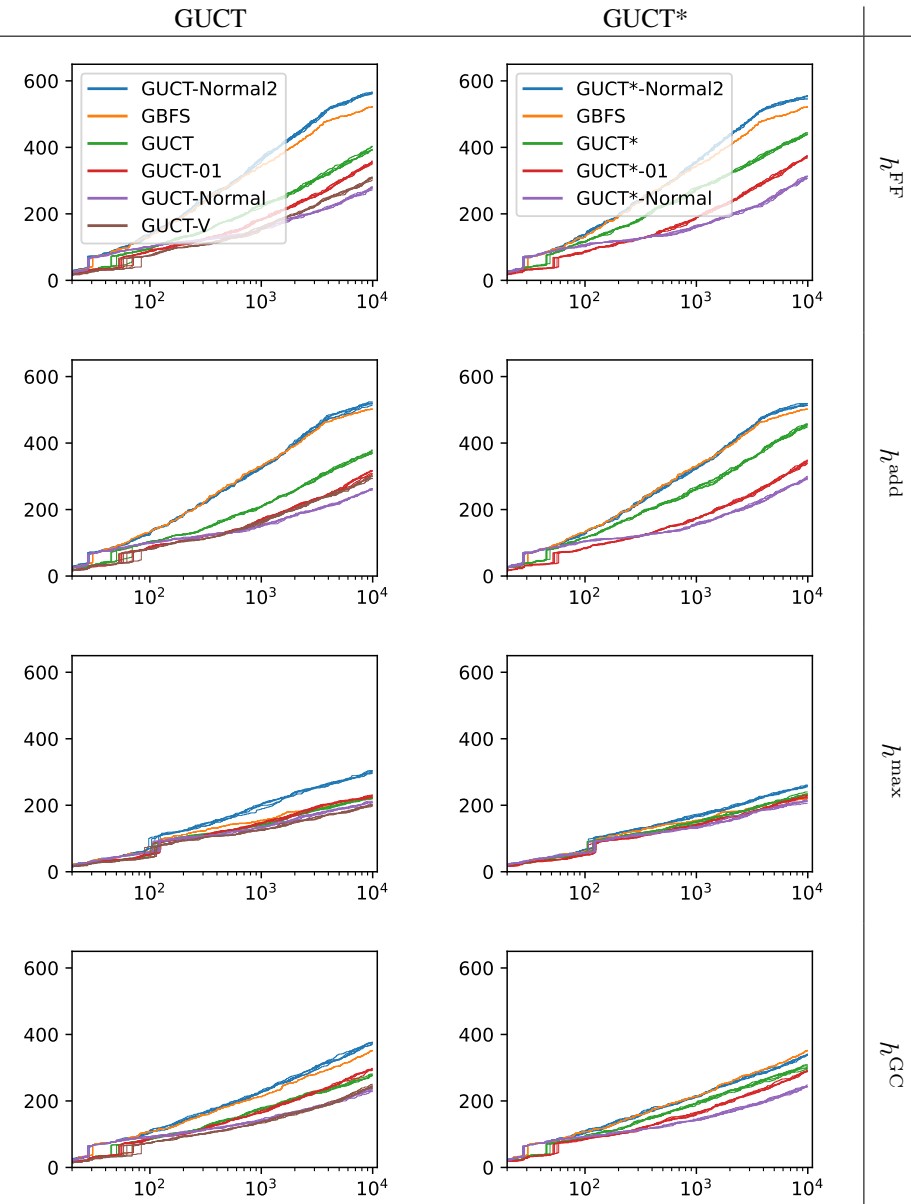

Figure S1: The cumulative histogram of the number of problem instances solved ($y$-axis) below a certain number of node evaluations ($x$-axis). Each line represents a random seed. In algorithms with an exploration coefficient hyperparameter, we use $c = 1.0$.

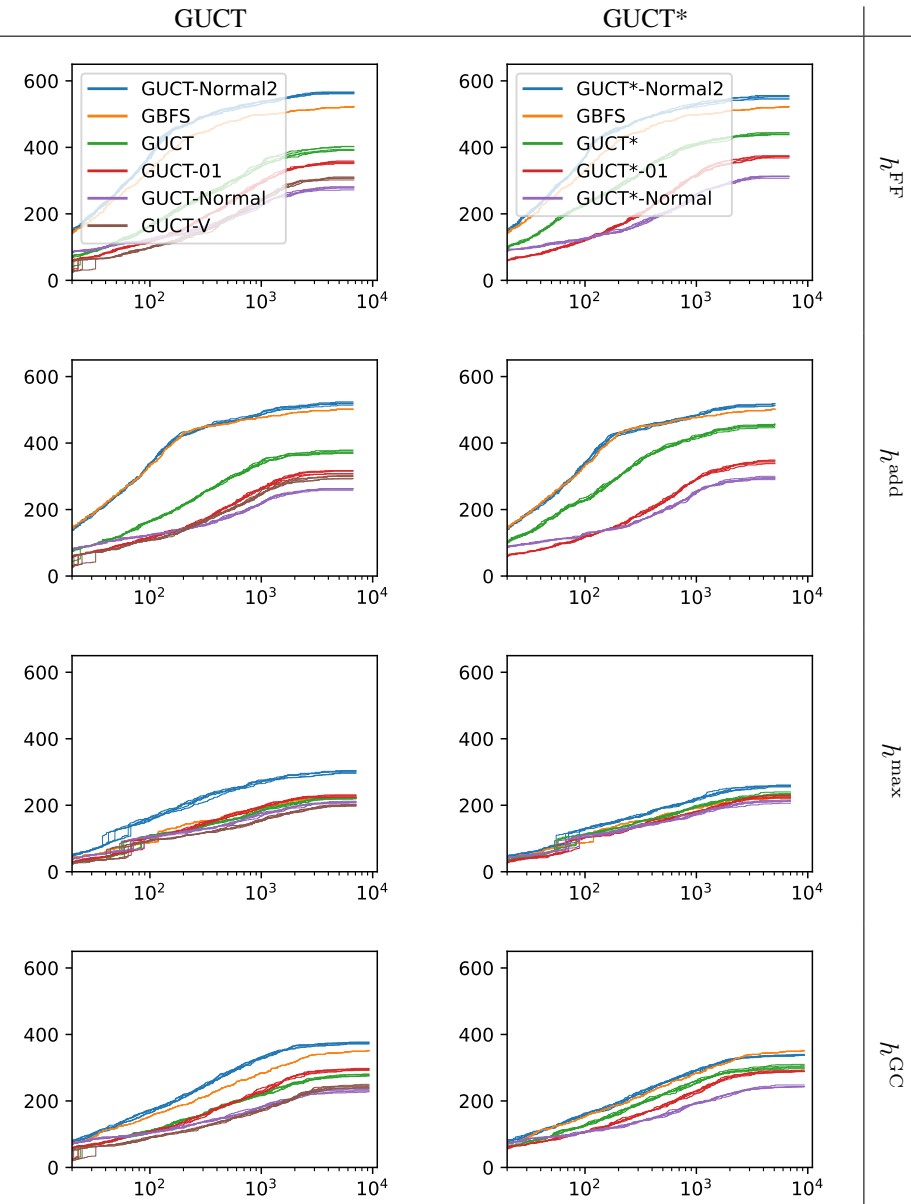

Figure S2: The cumulative histogram of the number of problem instances solved ($y$-axis) below a certain number of node expansions ($x$-axis). Each line represents a random seed. In algorithms with an exploration coefficient hyperparameter, we use $c = 1.0$.

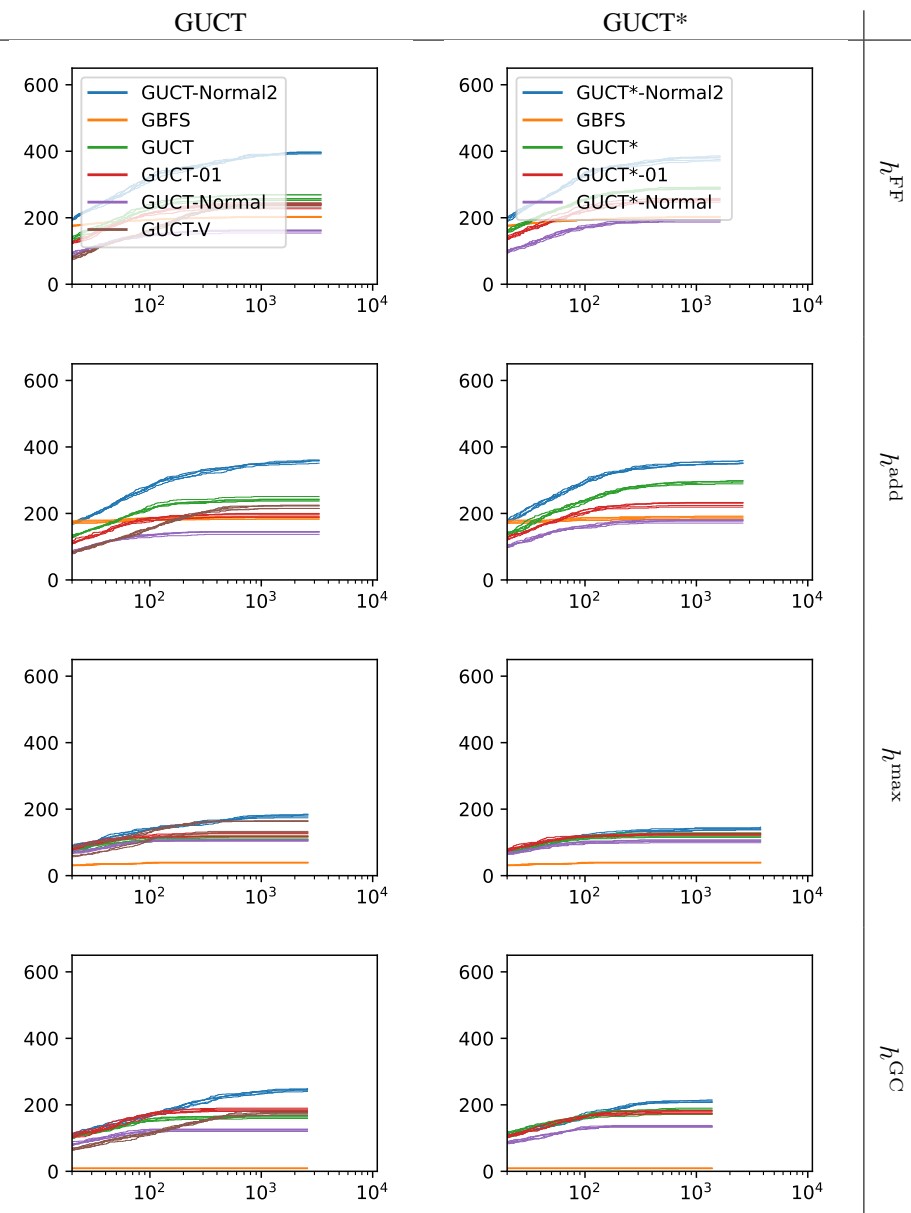

Figure S3: The cumulative histogram of the number of problem instances solved ($y$-axis) below a certain time elapsed ($x$-axis). Each line represents a random seed. In algorithms with an exploration coefficient hyperparameter, we use $c = 1.0$.

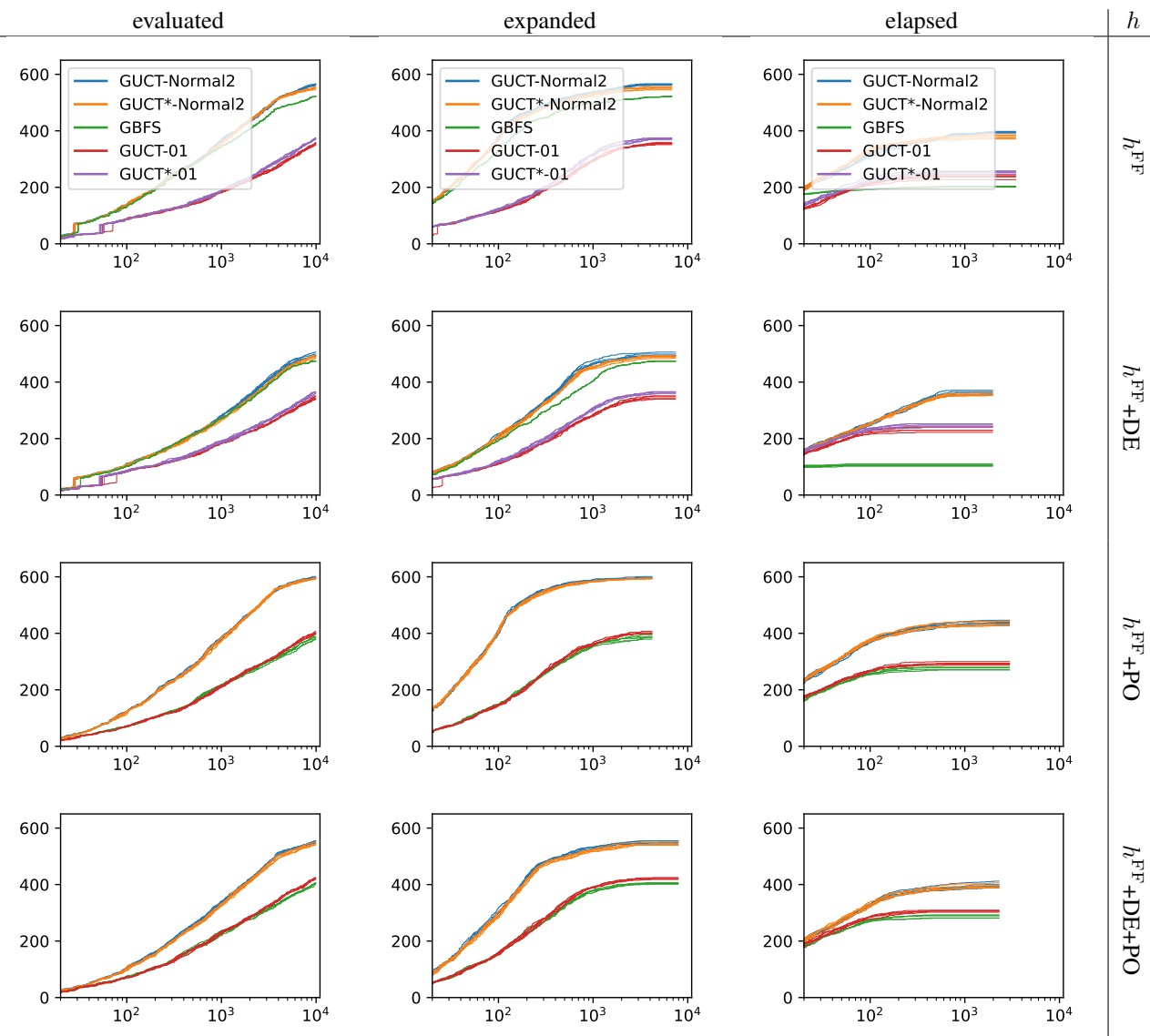

Figure S4: The cumulative histogram of the number of problem instances solved ($y$-axis) below a certain evaluations/expansions/runtime ($x$-axis). Each line represents a random seed. In algorithms with an exploration coefficient hyperparameter, we use $c = 1.0$.

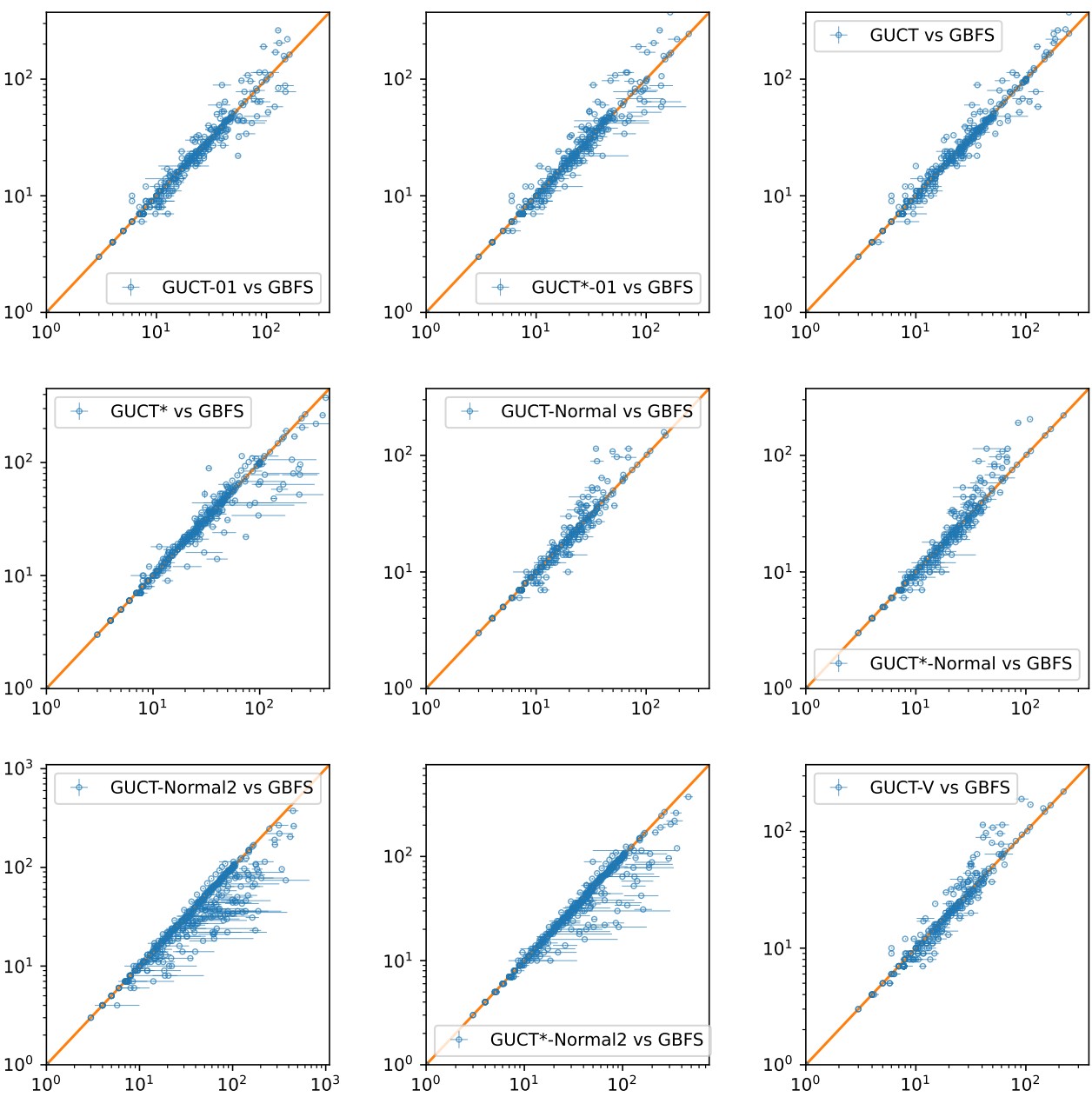

Figure S5: Comparing the length of solutions found by GUCT-based algorithms ($x$-axis) against those by the baseline GBFS ($y$-axis) using $h^{\text{FF}}$.

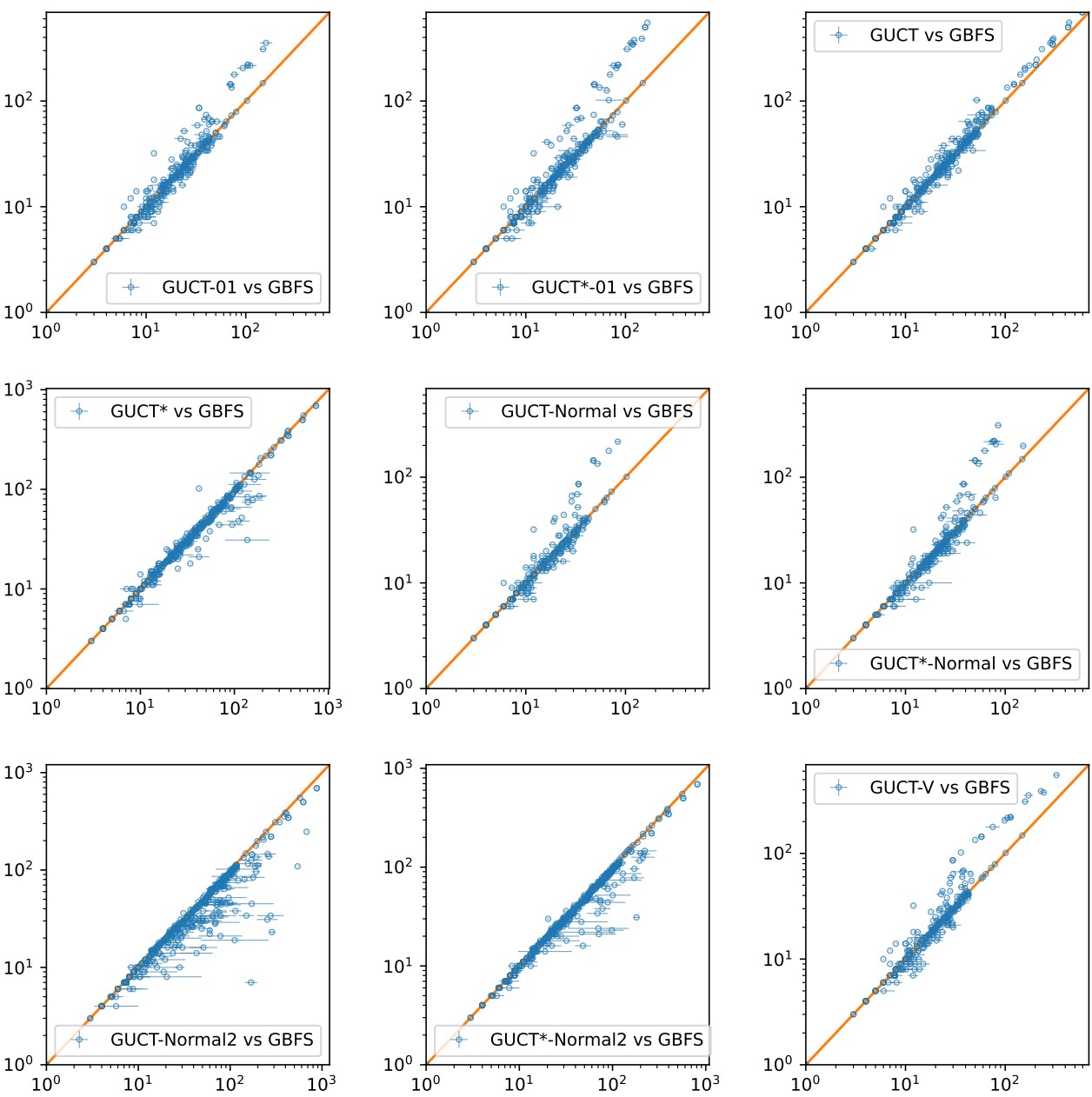

Figure S6: Comparing the length of solutions found by GUCT-based algorithms ($x$-axis) against those by the baseline GBFS ($y$-axis) using $h^{\mathrm{add}}$.

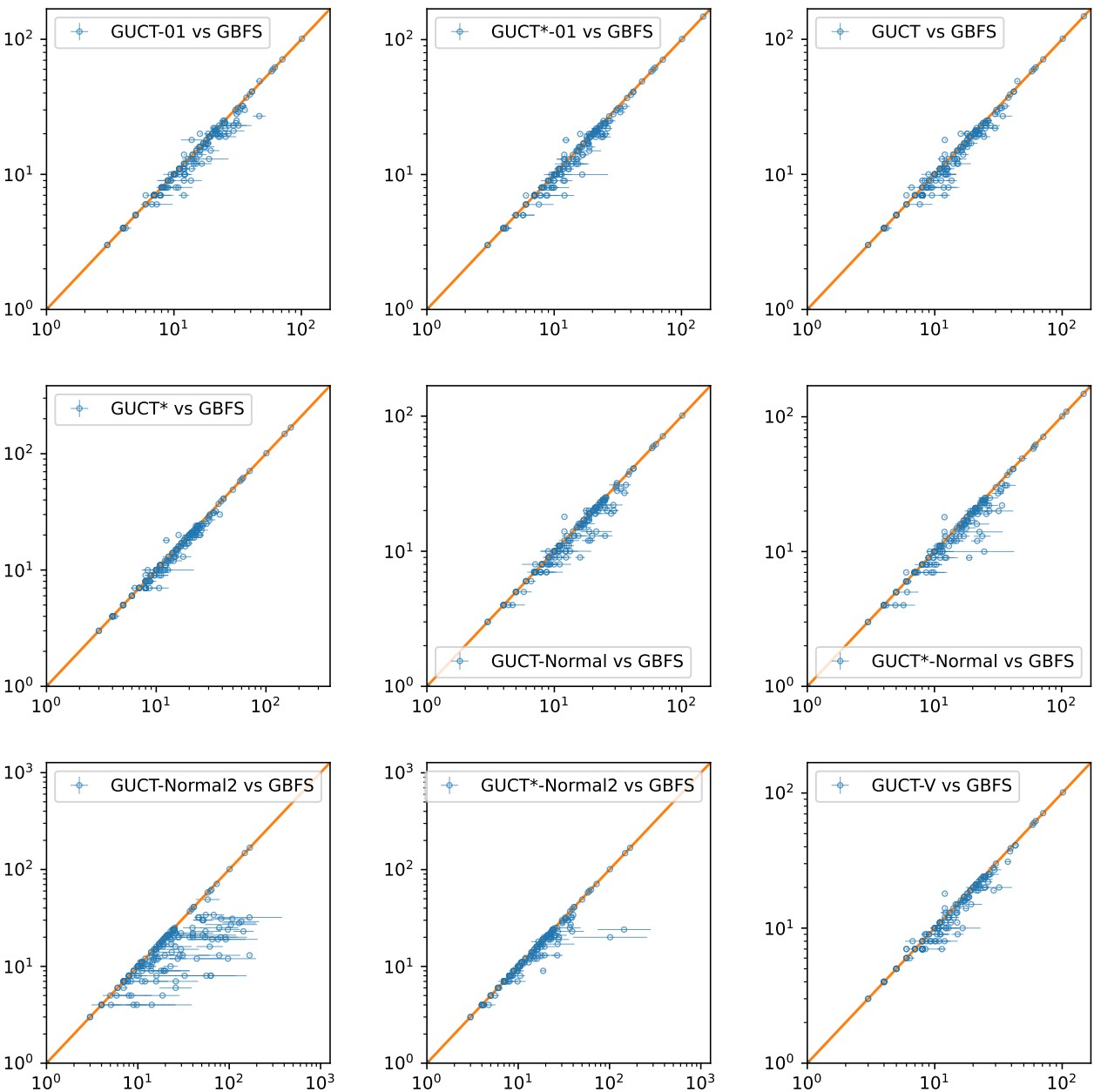

Figure S7: Comparing the length of solutions found by GUCT-based algorithms ($x$-axis) against those by the baseline GBFS ($y$-axis) using $h^{\mathrm{max}}$.

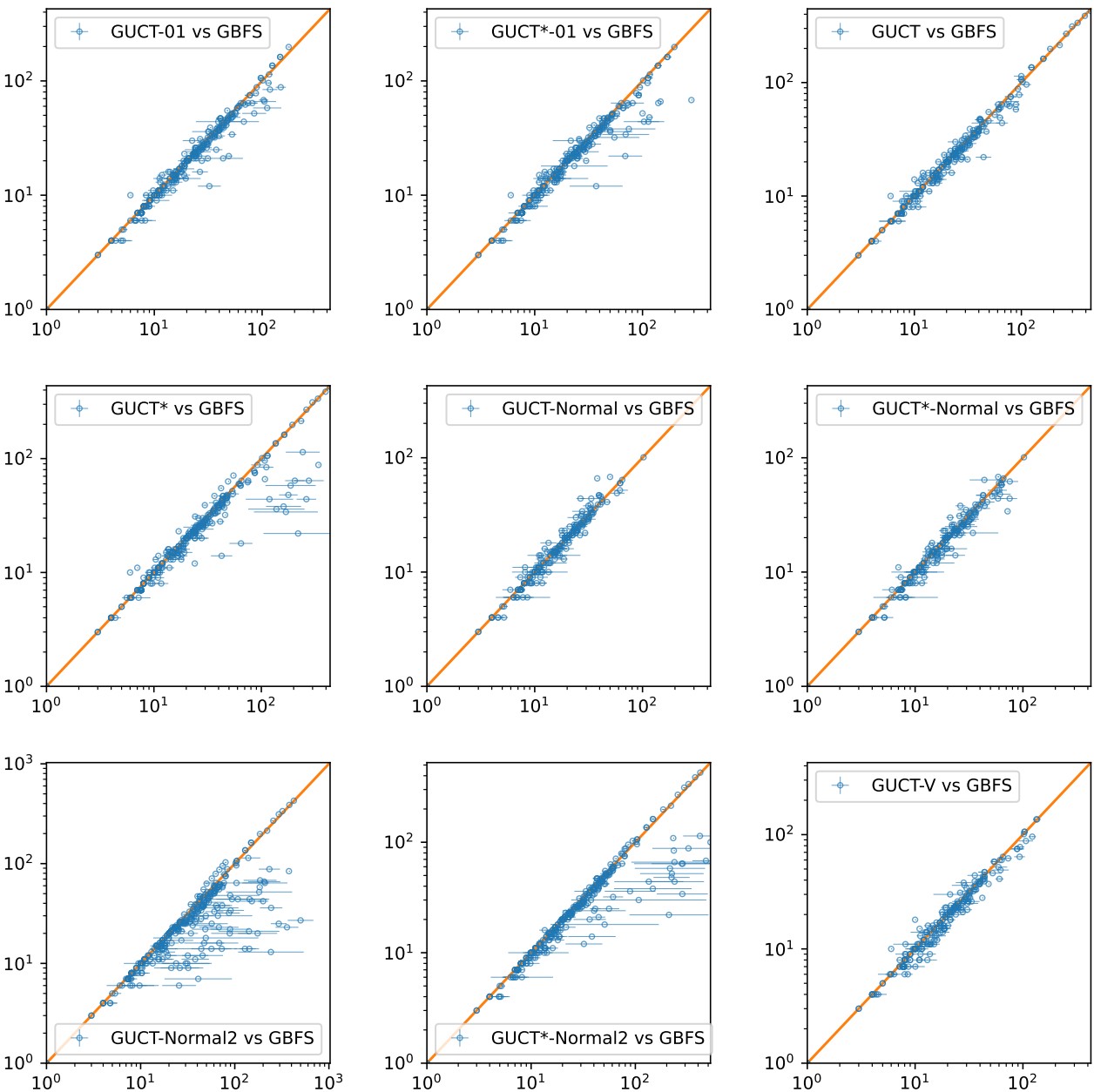

Figure S8: Comparing the length of solutions found by GUCT-based algorithms ($x$-axis) against those by the baseline GBFS ($y$-axis) using $h^{GC}$.