# OpenReview forum: "Scale-Adaptive Balancing of Exploration and Exploitation in Classical Planning"
_icaps-conference.org/ICAPS/2023/Workshop/HSDIP — ICAPS HSDIP 2023_

### Official Review · Reviewer_Fh9k · 2023-04-14
**Interesting approach to make MCTS work in classical planning with some issues in detail (novelty, omited baseline comparisons)**

**Rating:** 6
**Confidence:** 4

**Review:**

The paper investigates the use of variance-aware evaluation functions to guide MCTS for classical planning. The authors observe that UCT with its underlying UCB1 evaluation function has some downfalls in the classical planning setting. In particular, it requires a prior plan length bound for aligning UCB1's exploration term to the scale of the exploitation term. UCB1-Normal is a known but less popular variant of the UCB1 evaluation function designed for supporting multi-armed bandit problems with a priori unknown reward scales, and hence offers a remedy to this situation. It weights the exploration term by the variance of the corresponding rewards sampled so far, which (a) can be seen as a dynamic assessment of the respective arm's reward scale (as per the samples), and (b) dynamically weights the exploration term according to the reward uncertainty in the respective search subtree. The authors present a slightly modified variant, baptized UCB1-Normal2. They demonstrate experimentally that both variance-aware UCB1 forms outperform previous UCT implementations as well as the more traditional GBFS algorithm. Furthermore, the authors prove that their new UCB1-Normal2 variant guarantees a polynomial regret bound.

The topic of the paper fits well the workshop and furthermore it provides some interesting insights into the application of MCTS for classical planning. The first half of the paper is very well written, and offers a comprehensive overview of heuristic search and MCTS algorithms from a classical planning perspective, and the different variants of the UCB1 action-choice function known from multi-armed problems. Unfortunately, the quality of the paper drops significantly in the second and actually main half. The background material ends on the second column of 5th page, creating an imbalance between prior and novel results. The algorithm contributions are currently mixed with the presentation of techniques from prior work, which makes it difficult to assess what is novel and what not. Moreover, the proof of the main theorem (Theorem 2), spanning 3 pages of appendix, is completely underrepresented in the main text. I'd expect at least the main arguments to be summarized in a proof sketch format. The experimental evaluation considers the essential algorithm comparisons. But, the provided explanations are not really sufficient to connect to the theoretical assessment (see details below). Moreover, state-of-the-art algorithm configurations are not considered (preferred operators, deferred heuristic evaluation). All in all, the paper conveys some for the workshop interesting findings. But, the amount of technical issues places it rather the borderline of accept.

The authors criticize at various places the ``ad-hoc'' decisions made by Schulte and Keller (SOCS 2014) in their prior work on MCTS for classical planning. What is so ad-hoc about them? In particular, preferred operators has established itself as one of the most impactful heuristic search guidance mechanisms for satisficing planning. If a search algorithm A outperforms an algorithm B with this technique enabled, then this just demonstrates that A does make much better use of the provided information. This observation doesn't change only because A performs worse than B when this technique is disabled. Seeing the impact of preferred operators on Schulte and Keller's MCTS implementation, it absolutely must be considered in the experiments (even if just in addition to the existing comparisons). It is of course true that preferred operators don't preserve theoretical properties. But, given that the authors focus on satisficing planning anyway, this is not a valid counterargument.

Furthermore, the authors claim to provide a theoretical assessment of MCTS for classical planning. I couldn't find it. Is it the regret bound? What impact does this result have when applying the algorithm to classical planning? A guarantee on plan quality? Empirically, this doesn't seem to be the case. In fact, the found plans are actually length-wise worse than the ones found by Schulter and Keller's MCTS method. How does the latter observation actually connect to the theoretical analysis?

Fixing these two points would turn the paper into a, in my opinion, good workshop addition. Given that there is still some space left, you can include some additional statistics for the empirical evaluation. In particular, Figures 1, 2 provide only a summary of the results. But per-domain results are often much more informative. Actually, can you say anything about that? Is the picture the same, or are there any domains where some of the algorithms jump out of the general pattern?

Additional, smaller, comments:

- I could not follow the discussion in the second paragraph of section 4.2. The plan length shouldn't matter much in terms of the overall UCB1-Normal value given that plan length affects both the mean (exploitation term) as well as the variance (scaling the exploration term). That's the whole point with UCB1-Normal, you implicitly enforce the same scale to both exploitation and exploration terms. A large variance is just an indicator for poor heuristic estimates. In your example, why is the standard deviation necessarily 10^4 if the plan length is 100?

- In section 3.1, the discussion starting from below equation (1) to the paragraph below equation (2) is redundant and actually, in my view, does not make any sense. As mentioned in my summary above, to apply UCB1 to a non-unit reward setting, one needs to balance the exploitation and exploration terms. Schulte and Keller achieve exactly this by normalizing the exploitation term to one (equation 1), which obviously is equivalent to scaling the exploration term by the inverse factor.

- The figures are hardly readable as, e.g., lines are hard to distinguish. Moreover, figure 1 seems to be out of place. You can put some highlights of the empirical results into the introduction even without moving the figure there. You should move the figure to where it is actually being discussed.

---

> ### Author Response · Authors · 2023-05-01
> **Response to Fh9k**
>
> > both variance-aware UCB1 forms outperform &#x2026;
>
> UCB1-Normal (Auer 2002) + MCTS performed worst of all. See Fig1, where for $h^{FF}, h^{add}, h^{max}$, GUCT-Normal is the worst variant, and GUCT\*-Normal is the second worst variant. The performance ordering is
>
>     UCB1-Normal2 > GBFS > UCB1 == UCB1-01 > UCB1-Normal.
>
> As we note after Thm1, this is due to the conjectures made by UCB1-Normal's logarithmic regret proof (Appendix B of Auer et al. (2002), page 254), which are not guaranteed to hold.
>
> > imbalance between prior and novel results.
>
> We believe this background is necessary for understanding the problem we address and motivating our solution.
>
> > difficult to assess what is novel and what not.
>
> The key algorithmic contribution is UCB1-Normal2; everything else is from prior work.
>
> > the proof &#x2026; is completely underrepresented in the main text. &#x2026; proof sketch
>
> The proof is relatively easy to understand in terms of bandit theory, but the classical planning community is unlikely to have this background.
>
> We will attempt to provide a proof sketch in the revision, though note that our proof in S3&#x2013;S4 is already concise as we assume Hoeffding's inequality and skipped steps shared by UCB1's proof.
>
> > preferred operators
>
> As preferred operators and lazy evaluations lack theoretical explanations, experiments involving them provide limited theoretical insight, which is our focus in this work.
>
> > claim to provide a theoretical assessment of MCTS for classical planning.
>
> We do not claim to provide such an assessment. MCTS is a meta-algorithm, only becoming concrete when combined with a specific action selection, such as a specific MAB.
>
> We do claim our analysis
>
> > deepens our theoretical understanding of heuristic search in general.
>
> We see many forward search algorithms (A\*, GBFS, &#x2026;) as a process of collecting a dataset (of leaf nodes) for each internal node, and exploring the environment based on updated estimates obtained from the collected data.
>
> > Is it the regret bound? What impact &#x2026; to classical planning? A guarantee on plan quality? &#x2026; How does &#x2026; connect to the theoretical analysis?
>
> The regret bound differs from guarantees typically discussed in the planning community, and instead guarantees the algorithm will make a "good" local decision at each internal node in a certain quantifiable sense. Analyzing the theoretical relationship between such locally optimal decision making and the global outcome of the search is an interesting avenue for future work.
>
> One purpose of satisficing/agile search is to prioritize finding solutions faster over finding lower cost solutions. Our analysis shows that GUCT-Normal2 dominates GUCT with all exploration coefficients tested under almost all expansion limits tested (Figure 2), and it is not unexpected to see solution cost suffering as a result.
>
> We have a longer answer which we will add if we are allowed to reply more later.
>
> > per-domain results
>
> We have scatter plots of node expansions, evaluations, solution-found, and runtime, comparing every pair of algorithms tested. We can add them in the revised appendix.
>
> > the second paragraph of section 4.2.
>
> In this paragraph, we argue that GUCT-Normal2's outperformance of GUCT-Normal provides evidence in support of our hypothesis, in the first paragraph after Theorem 2, that alpha is small and the variance large in UCB1-Normal2's polynomial regret bound, which we had noted would be necessary conditions for it to outperform UCB1-Normal's logarithmic regret bound in practice.
>
> > why is the standard deviation necessarily $10^4$ if the plan length is 100?
>
> Thanks; our mention of the variance here is a mistake and will be removed. Plan length 100 does not imply variance $10^4$ (stdev 100).
>
> > In section 3.1, the discussion &#x2026; does not make any sense.
>
> Answered in the general comment.
>
> > to apply UCB1 to a non-unit reward setting, one needs to balance &#x2026;
>
> It is theoretically justified to apply UCB1 to a non-unit reward setting by normalizing the exploitation term, given that *all arms* are known to have a bounded reward distribution *with the same bound*. Balancing (normalizing the exploitation term or scaling the exploration term) does not automatically guarantee the performance promised by UCB1.
>
> In adversarial games, the support is known: the maximum is 1 (win), and the minimum 0 (loss), thus the constant $c=1-0=1$. This applies to any successor node in the action selection.
>
> The assumption does not hold among the heuristic values of successors in classical planning (where rewards = negative heuristic values), because there is no a priori known max/min of heuristic values in each successor's subtree. To obtain one, one must exhaustively expand all reachable states and enumerate all values.

---

### Official Review · Reviewer_cSkB · 2023-04-26

**Rating:** 8
**Confidence:** 3

**Review:**

The paper analyzses search algorithms for classical planning that are based on
MCTS from a statistical viewpoint. The existing approach Trial-based Heuristic
Tree Search (THTS) adds an exploration term based on UBC1 to the node
priority. Since UBC1 requires a bounded reward distribution like [0,1], it
normalizes based on a node siblings. This paper points out that this leads to
different normalizations in different parts of the search, and instead suggests
different exploration terms that normalize in such a way that all parts of the
search space use the same shared scale.
The experimental evaluation compares the original THTS (without additional
enhancements) as well as their variants with new exploration terms against
GBFS, and show that the original THTS performs worse than GBFS in terms of node
expansions, while the new variants can outperform GBFS.

The topic of the paper fits well to HSDIP, and the contribution is to the best
of my knowledge very novel, since it for the first time analyzes THTS from a
statistical viewpoint. The experimental evaluation is convincing and provides
some valuable insights; I especially liked the emphasis in Section 4.2 on why
Normal2 is a good choice despite worse theoretical guarantees. I also commend
the authors for clarifying from the beginning that they use an unoptimized code
base and can thus not claim state-of-the-art, and I think this is a sensible
choice since the paper's focus is on analyzing theoretical properties that can
be experimentally observed with node expansions.

The paper is in general structured well. It is uncommon to already present some
experimental results in the introduction, and I'm not sure if it's a good idea
since at this point it is harder to read the plots, but I don't have a strong
opinion. In terms of clarity, I especially liked the extensive background
section that did a great job of explaining MCTS concepts. However, section 3 in
comparison remained rather unclear to me, specifically subsection 3.1. The
sentence "However, this fails to account for the fact that the scale of each
arm is different" confused me since I was not sure what arm means here in terms
of the search space. I came to the conclusion that it means different areas of
the search space, e.g. in one expansion the heuristic values from the
successors might range from 2-8 and get normalized with 2->0 and 8->1, while in
another area the it ranges from 3-4 and thus we have 3->0 and 4->1.
Furthermore, I don't understand the purpose of equation 2 and I don't
understand what behavior is preserved. Is it a fix for equation 1? I at first
thought so because it "uses the same shared scale" which I though is what we
want; but it is not used anywhere in the experiments, and the sentence before
starts with "Consider" which to me rather suggests that you want to show an
example on why equation 1 is bad.

Despite the clarity issues in section 3 I think that this is a strong paper
that provides valuable insight in MCTS-like classical planning algorithms and
clearly recommend to accept it. I would however strongly recommend to revise
section 3 to make it clearer.


Minor comments:
 - Figures: I would refrain from interpreting the results of figures in the
 caption, and only use the caption to describe what we see in the figure.
 - Figure 1 is hard to read because the subfigures are rather small, there are
 a lot of overlapping lines and some colors are very similar. (But
 unfortunately I don't really know how to improve it other than increasing the
 size.)
 - Introduction, last paragraph. "(See our ... in Sec.4). -> period inside the
 brackets since the brackets contain a complete sentence, e.g. "(See our ... in
 Sec.4.)
 - Section 2.4, end of third paragraph. "It also implicitly detects a cycle".
 How? Do we assume undirected search spaces?
 - Section 3, paragraph before 3.1: "Note that the sample size |L(n)| is
 sometimes inaccurately called a "visitation counter". Why is it inaccurate?
 - Section 4, second paragraph: "Fast-Downward" -> "Fast Downward". But Fast
 Downward does no longer include the IPC benchmark set, they are now separately
 hosted under https://github.com/aibasel/downward-benchmarks.
 - Figure 3: I don't understand how the plot works. Does the size of the cross
 indicate the range of node expansions? What are the circles? What do the
 colors mean? Why is there no legend?
 - Section 4.1, second paragraph: Why Is Fig.3 referenced before Fig.2? I would
 recommend to number the figures in order of text appearance.
 - last paragraph of Section 4.2: I am not fully convinced that GUCT-Normal2
 outperofrms GUCT for all values of c. When reaching the node expansion limit,
 c=0.1 catches up to GUCT-Normal 2 and sometimes even overtakes it
 (specifically, one of the c=0.1 lines has overall highest coverage at the
 end). This might suggest that given more resources, c=0.1 might actually be
 the better choice.
 - References: page numbers are missing in many references
 - References: Pyperplan misses a URL or DOI. Their github repository provides
 a bibentry for referencing, I suggest to use this.


Questions to the authors:
1) Is my intuition on what an "arm" in section 3.1 is correct?
2) Can you elaborate on the purpose of equation 2? Is it a fix for equation 1?
3) I was at first suspicious that reporting on expansions rather than runtime
might hide the fact that each expansion is more expensive. But I guess this is
not the case here since the different algorithms only differ in the exploration
term, which should not be hard to compute. Is this assessment correct?

---

> ### Author Response · Authors · 2023-05-01
> **Response to cSkB**
>
> > This paper points out that this leads to different normalizations in different parts of the search,
>
> We pointed out that, during the selection phase of MCTS, the distribution of heuristic values in the subtree of each successor has a different scale, while UCB1 and UCB1-01 pretend the scale is the same among successors.
>
> > and instead suggests different exploration terms that normalize in such a way that all parts of the search space use the same shared scale.
>
> On the contrary: UCB1-Normal and UCB1-Normal2 do not normalize. They scale the exploration term by sigma, which varies among successor nodes and thus throughout the tree.
>
> > The sentence "However, &#x2026; " confused me since I was not sure what arm means &#x2026;
>
> In its action selection phase, MCTS repeatedly chooses a successor node, starting from the root of the search tree. We use a bandit algorithm for this purpose, so the arm corresponds to an action and its corresponding successor node.
>
> > e.g. in one expansion the heuristic values from the successors might range from 2-8 and &#x2026;
>
> The correct example would be: a node $n$ has two children $n_1$ and $n_2$, each with its own children (grandchildren of $n$). The children of $n_1$ have heuristic values ranging $[2,8]$, and the children of $n_2$ have $[3,4]$. As the subtrees rooted at $n_1$ and $n_2$ grow, and the set of leaf nodes changes and grows, these ranges are updated.
>
> > &#x2026; purpose of equation 2 and &#x2026; what behavior is preserved.
>
> Answered in the general comment.
>
> > Figure 1 is hard to read &#x2026;
>
> We will improve the figure quality in the revision to better distinguish the lines.
>
> > "It also implicitly detects a cycle"
>
> The lock mechanism in THTS acts as duplicate detection in Dijkstra/A\*/GBFS.
>
> > "visitation counter". Why is it inaccurate?
>
> The visitation counter is the number of times a node was selected by the action selection, and differs from $|L(n)|$ up to branching factor $B$ in the following way. Imagine a node $g$ (grandparent), $p$ (parent), and $p$'s children. Assume $p$ has been selected by $g$ only once, therefore $p$'s $B$ children are all leaves, and $p$'s visitation counter is 1 (from its expansion). In the next expansion, when the action selection reaches $g$ and selects from $p$'s siblings, the bandit algorithm treats $p$ as having only 1 sample, which is obviously incorrect because the number of samples is the number of leaf nodes in the subtree $|L(p)|$, which is $B$.
>
> The "number of samples" interpretation thus aligns more closely with Bandit theory, which handles the reward dataset collected from each arm.
>
> > <https://github.com/aibasel/downward-benchmarks>.
>
> We were indeed referring to this repository, and we will clarify the citation.
>
> > Figure 3
>
> The size of the cross is the errorbar computed from standard error (standard deviation divided by \sqrt{n}) among random seeds. The size of the circle is the same for all points and does not imply anything.
>
> As described in the captions, the purpose of these scatter plots is to show that there is no significant difference between two configurations compared in each axis (UCB vs UCB-01). Each color corresponds to a domain, but UCB1 and UCB1-01 behave the same in all domains, so we will convert all colors to black for camera-ready to avoid confusion.
>
> > c=0.1 catches up to GUCT-Normal 2 &#x2026;
>
> To limit experiment length, we terminated a given run when it exceeded either a 10,000 node expansion limit or a 15 minute running time limit (disjunction of termination criteria).
>
> However, this may have affected data at higher expansion limits, as suggested by the flattened curve at the rightmost end of the plot.
>
> We are now (a) running experiments that terminate runs only when the **conjunction** of the termination criteria becomes true, and then (b), in each plot, filtering data by a single termination criteria focused by the plot, to eliminate any confounding effect from the resource limits.
>
> > runtime
>
> Our preliminary runtime analysis in Appendix S7 shows that GUCT-Normal2 solves more instances than priority queue-based GBFS under a given running time; we can include additional plots from the appendix in the revision if preferred.

---

### Author Response · Authors · 2023-05-01
**Overall response**

Thank you for your thoughtful comments. We will update the paper to address all minor points. In particular, both reviewers commented about the quality of the figures, and we will improve them in the revision.

We first want to clarify that we use the term "normalize" in a strict sense. "Normalizing a value" means **dividing** a value by the scale so that the result has a unit scale (scale = 1).

We do **not** call arbitrary multiplication with the scale "normalization," as the result is not guaranteed to have a unit scale; we would instead call it "scaling."

> (Reviewer cSkB) the purpose of equation 2 / what behavior is preserved.

> (Reviewer Fh9k) In section 3.1, the discussion &#x2026; is redundant

We would like to clarify the discussion of Eq.1-2 for both reviewers.

GUCT-01's action selection selects the successor node with the smallest value of $f_{guct-01}$ (Eq.1). Eq.1 is mathematically equivalent to Eq.2. Eq.2 is the same as the original UCB1 $f_{guct}$ with hyperparameter $c$ replaced with $c(M-m)$. Therefore, GUCT (MCTS + UCB1) with hyperparameter $c(M-m)$ and GUCT-01 (MCTS + UCB1-01) with hyperparameter $c$ select the same successor.

The discussion emphasizes that the modification by Schulte and Keller results in the same UCB1 algorithm with a different hyperparameter, thus, just like UCB1, UCB1-01 equally fails to account for scale difference among the successors during action selection. This explains why the performance with and without normalization is the same on average (Figure 3, Sec.4.1).

We can't say the node expansion order is the same in UCB1 and UCB1-01 because each internal node at a different depth has different M and m. However, Figure 3 suggests that the effect of this difference is minimal.

We will post separate replies to the remainder of each initial review momentarily.

---

> ### Comment · Reviewer_cSkB · 2023-05-02
> **Clarification on what correct normalization is**
>
> Thank you very much for your response, I understand the purpose of Equation 2 much better now. Small caveat: Equation 2 also translates everything by m, but I assume this is not a problem: since the m is the same between all siblings, it doesn't affect the ranking either.
>
> What is still unclear to me is how proper normalization looks like. Consider the following small example on a unit-cost task: We have initial state $s_0$ with children $s_1$ and $s_2$, while $s_1$ has children $s_{1,1}$ and $s_{1,2}$, and $s_2$ has childern $s_{2,1}$ and $s_{2,2}$. Furthermore we know $h(s_{1,1}) = 3$, $h(s_{1,2}) = 4$, $h(s_{2,1}) = 1$ and $h(s_{2,2}) = 12$. This leads to $h_{UTC}(s_1) = 4.5$, $h_{UTC}(s_2) = 7.5$ and $h_{UTC}(s_0) = 7$ if I'm not mistaken. Now, when computing $f(s_1)$ and $f(s_2)$, do we want to normalize in such a way that everything is scaled from 1 to 12, or do we want to normalize $s_1$ with respect to (3,4) (or maybe (4,5), taking the cost into account?) and $s_2$ with respect to (1,12) (or (2,13))? I assume the latter, hence "adaptive scale". Could we not also achieve this by selecting m and M differently, i.e. always with respect to the nodes' children rather than siblings?

---

> > ### Author Response · Authors · 2023-05-03
> > **Response to "Clarification on what correct normalization is"**
> >
> > Thank you for your follow up questions.
> >
> > > Equation 2 also translates everything by m, but I assume this is not a problem: since the m is the same between all siblings, it doesn't affect the ranking either.
> >
> > Yes, this is correct.
> >
> > > Could we not also achieve [better normalization] by selecting m and M differently, i.e. always with respect to the nodes' children rather than siblings?
> >
> > That is an interesting idea for future work. The challenge would be to make it into a proper bandit and to analyze the regret theoretically.

---

### Decision · Program_Chairs · 2023-05-05

**Decision:**

Accept

**Comment:**

We are happy to announce that the paper is accepted for presentation at HSDIP.

Please make sure to address the reviewer comments and clarify the questions in the final version.